# The C-terminus of the multi-drug efflux pump EmrE prevents proton leak by gating transport

**Merissa Brousseau[1†], Da Teng[2†], Nathan E Thomas[1‡], Gregory A Voth[2], Katherine A Henzler-Wildman[1,3]\***

[1]Department of Biochemistry, University of Wisconsin-Madison, Madison, United States; [2]Department of Chemistry, Chicago Center for Theoretical Chemistry, Institute for Biophysical Dynamics, and The James Franck Institute, The University of Chicago, Chicago, United States; [3]Nuclear Magnetic Resonance Facility at Madison, University of Wisconsin-Madison, Madison, United States

**\*For correspondence:**
henzlerwildm@wisc.edu

[†]These authors contributed equally to this work

**Present address:** [‡]Labcorp Drug Development, Madison, United States

**Competing interest:** The authors declare that no competing interests exist.

## eLife Assessment

This study provides a **fundamental** analysis of the EmrE efflux pump, highlighting the role of the C-terminal domain in influencing uncoupled proton leak. The integration of biophysical techniques with molecular dynamics simulations offers **solid** support for the key findings and adds substantial evidence toward a definitive understanding of EmrE transport mechanism.

**Abstract** The model multi-drug efflux pump from *Escherichia coli*, EmrE, can perform multiple types of transport leading to different biological outcomes, conferring resistance to some drug substrates and enhancing susceptibility to others. While transporters have traditionally been classified as antiporters, symporters, or uniporters, there is growing recognition that some transporters may exhibit mixed modalities. This raises new questions about their regulation and mechanism. Here, we show that the C-terminal tail of EmrE acts as a secondary gate, preventing proton leak in the absence of drug. Substrate binding unlocks this gate, allowing transport to proceed. Truncation of the C-terminal tail (Δ107-EmrE) leads to altered pH regulation of alternating access, an important kinetic step in the transport cycle, as measured by NMR. Δ107-EmrE has increased proton leak in proteoliposomes, and bacteria expressing this mutant have reduced growth. Molecular dynamics simulations of Δ107-EmrE show the formation of a water wire from the open face of the transporter to the primary binding site in the core, facilitating proton leak. In WT-EmrE, the C-terminal tail forms specific interactions that block the formation of the water wire. Together, these data strongly support the C-terminus of EmrE acting as a secondary gate that regulates access to the primary binding site.

## Introduction

Antibiotic resistance may be mediated by several mechanisms, including active export of drugs by promiscuous multi-drug efflux pumps (*Munita and Arias, 2001*). Among these transporters, the small multi-drug resistance (SMR) family has been found throughout the bacterial kingdom and exhibits particularly promiscuous substrate profiles (*Brown and Skurray, 2001*; *Pérez-Varela et al., 2019*; *Schuldiner, 2009*). The most well-studied SMR transporter is the *Escherichia coli* protein EmrE, which confers resistance to a broad array of toxic polyaromatic cations and quaternary ammonium compounds through secondary active transport (*Yerushalmi and Schuldiner, 2000a*). These transporters couple

**Figure 1.** Model of coupled antiport and uncoupled proton leak through EmrE. (**A**) All of the drug- and proton-bound states that are reasonably populated at near-physiological temperature and pH and the transitions between these states observed by NMR lead to a model for EmrE transport that allows for both coupled antiport (orange) and proton leak (red solid line). (**B**) In WT-EmrE, the C-terminal tail on the open face acts as a secondary gate (top), minimizing proton leak in the absence of substrate. Truncation of EmrE in Δ107-EmrE removes this gate (bottom). The drug binding to a secondary binding site near the tail opens the gate (top,right), allowing proton exit from the primary binding site near E14, and drug to progress to the primary binding site at E14. This leads to either coupled antiport (A, orange) as shown. If the substrate does not rapidly move into the primary binding site, only proton entry/exit occurs upon opening of the secondary gate, resulting in drug-gated proton leak (A, red dashed line). Truncation of the C-terminal tail in Δ107-EmrE (B,bottom) allows uncoupled proton leak in the absence of substrate (A, red solid line).

the energetically favorable import of protons down the proton motive force to drive active export of antibiotics and antiseptics (*Forrest et al., 2011*; *Boudker and Verdon, 2010*). As the archetype for the family of the smallest ion-coupled transporters, EmrE has become a model system for studying the molecular mechanism of proton-coupled transport and multi-drug efflux.

EmrE transport is electrogenic for tetraphenylphosphonium ($TPP^+$) and electroneutral for methyl viologen ($MV^{2+}$) (*Rotem and Schuldiner, 2004*), consistent with a 2$H^+$:1 drug antiport stoichiometry (*Yerushalmi and Schuldiner, 2000a*). Early mechanistic models focused on the minimal set of states and transitions necessary for such stoichiometric antiport. More recently, NMR studies of EmrE protonation and alternating access showed that many more states and transitions have populations and rates that are not insignificant at near-physiological pH and temperature (*Robinson et al., 2017*). Inclusion of these states and transitions in the mechanistic model provides pathways that allow for alternative transport activity, including symport, drug uniport, and proton uniport (leak) (*Figure 1*). In this model, different environmental conditions (pH) or small molecule substrates that shift the relative rates of different microscopic steps can alter the dominant transport behavior (*Robinson et al., 2017*;

*Hussey et al., 2020*). This has been confirmed experimentally: a small molecule substrate, harmane, triggers uncontrolled proton leak through EmrE to an extent that is detrimental to *E. coli* growth and NADH production (*Spreacker et al., 2022*). However, the question of how this transporter avoids catastrophic leaks remains unanswered.

Direct measurements of proton release upon drug binding showed that drug-induced deprotonation occurs at the C-terminal histidine (H110) in addition to the essential glutamate-14 residues that define the primary binding site for drug and proton (*Thomas et al., 2018*). Additionally, the C-terminus on one protomer in the homodimer is highly sensitive to the identity of the drug bound in the primary site (*Morrison and Henzler-Wildman, 2014*). Early solid-state $^{31}$P NMR experiments suggested a second, lower affinity TPP$^+$-binding site near the acidic loop residues E25 and D84 (*Glaubitz et al., 2000*), which are likely to be in close spatial proximity to the C-terminal tail in this small transporter. Together, these data led us to propose a secondary gating model where the C-terminal tail prevents proton release until drug binding at a peripheral site on the transporter surface displaces the tail (*Thomas et al., 2018*).

Unfortunately, none of the available EmrE structures provide high-resolution data on the conformation of the C-terminal tail and adjacent loop regions. Early cryo-electron microscopy and crystal structures revealed the unique asymmetric arrangement of the transmembrane helices and antiparallel topology of the EmrE homodimer, but had very low resolution and limited density in the loops and tails (*Fleishman et al., 2006*; *Chen et al., 2007*). Recent, higher-resolution crystal structures and NMR structures have provided more precision on substrate binding within the transport pore (*Kermani et al., 2022*; *Shcherbakov et al., 2022*; *Shcherbakov et al., 2021*). However, the crystal structures used a monobody that required mutation of three residues in the TM1–TM2 loop (E25N, W31I, and V34M), including E25, and there is limited or missing density for other loops on the open face of the transporter and the C-terminal tail after residue 104. In the NMR structures, distance restraints are primarily substrate–protein distances within the transmembrane helices lining the primary binding site, and only chemical-shift-derived backbone torsion angles restrain the loops and tail (*Shcherbakov et al., 2022*; *Shcherbakov et al., 2021*). The most recent NMR structures used a loop mutant, L51I, that disrupts the gating mechanism, locking the transporter open, and again had limited restraints in the loops and C-terminal tail (*Li et al., 2024*). Thus, there is limited structural data for the C-terminal tail and loops, although these regions are functionally important in gating access to the central binding site defined by residue E14. In such cases, molecular dynamics (MD) has proven to be an excellent tool, and the only atomic resolution model of the C-terminal tail is from MD simulations (*Vermaas et al., 2018*).

Here, we use NMR, in vitro and in vivo biochemical assays, and MD simulations to characterize a C-terminal deletion mutant of EmrE truncated after residue 106, denoted Δ107-EmrE, to directly determine the regulatory role of the C-terminal tail. Comparisons of growth and resistance phenotypes, alternating access rates, and transport activities of Δ107- and WT-EmrE confirm the importance of the C-terminal tail in regulating tightly coupled antiport and minimizing proton leak. Simulations on these two systems also suggest differences in water structure and hydrogen bonding patterns when the C-terminus is truncated. Examination of interactions with the newly discovered substrate harmane, which triggers uncoupled proton leak as the dominant transport mode (*Spreacker et al., 2022*), suggests interactions between the tail and a secondary site on the protein may allow for allosteric regulation of gating, reconciling the free exchange model with minimal leak by the WT transporter in the absence of small molecule substrates. Further, MD simulations provided a possible secondary site and the structural basis for this regulation.

## Results

### EmrE is properly folded and functional when the C-terminal tail is truncated

We first assessed whether C-terminal tail truncation affected the ability of EmrE to confer resistance to toxic substrates, the well-established primary function of this transporter. Growth assays of MG1655-Δ*emrE E. coli* cells expressing WT-, Δ107-, or E14Q-EmrE show that all strains grow well in the absence of toxic compounds (*Figure 2B*). In these assays, uninduced leaky expression from a low copy number plasmid (p15 origin) with a pTrc promoter keeps transporter expression relatively low (*Spreacker*

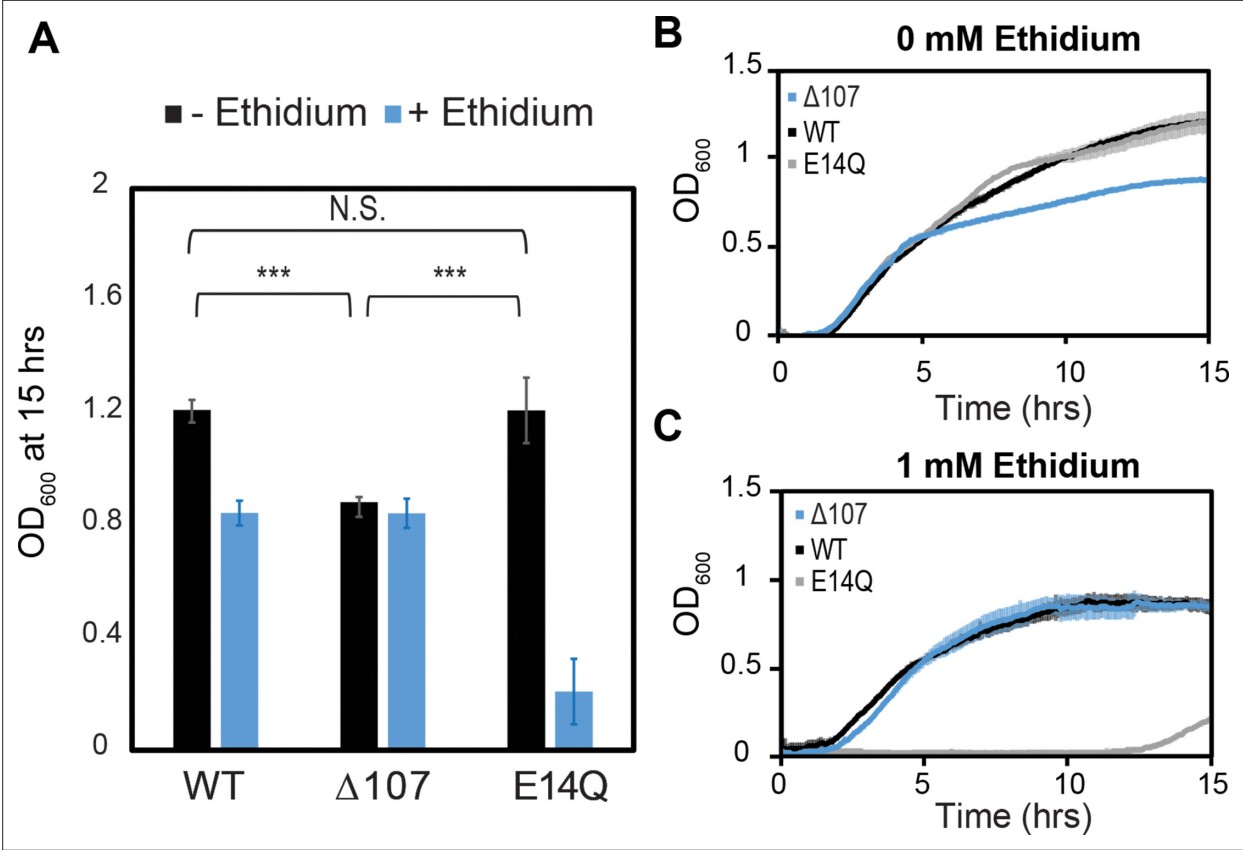

**Figure 2.** C-terminal tail truncation does not impair the ability of EmrE to confer resistance to toxic compounds. (**A–C**) WT-, E14Q-, or Δ107-EmrE was heterologously expressed in MG1655-Δ*emre E. coli* using a plasmid with p15 origin and pTrc promoter without induction to minimize any growth defect due to expression. In vivo growth assays were monitored by OD700 to allow consistent monitoring in the absence (**B**) or presence of (**C**) ethidium bromide. Growth at 15 hr. (**A**) shows identical growth for WT-EmrE and Δ107-EmrE in the presence of ethidium, while E14Q-EmrE is severely impaired (**A, B**). There is a 20% reduction in growth for Δ107-EmrE relative to WT-EmrE or non-functional EmrE (p < 0.001), but this does not prevent the mutant from transporting ethidium out of the cell and thus conferring resistance (**A, C**). The error bars show the standard deviation across six replicates (two biological replicates with three technical replicates each). All p-values were calculated from a two-sided *t*-test. ***p < 0.001.

*et al., 2022*). In the presence of ethidium bromide, a substrate commonly used to assess the activity of EmrE and other multi-drug efflux pumps, a functional transporter is required for survival (*Figure 2C*). This confirms the known resistance activity of WT-EmrE, that E14Q-EmrE is non-functional, and that Δ107-EmrE is properly expressed to the inner membrane, folded and functionally able to confer resistance to toxic compounds in a manner comparable to the WT transporter.

### Truncation of the C-terminal tail enhances proton leak through EmrE

There is a small but reproducible growth defect for cells expressing Δ107-EmrE in the absence of exogenous substrate (*Figure 2A, B*). This defect becomes apparent around 5 hr, the point at which available fermentable sugars in LB media are depleted, increasing dependence on the proton motive force for energy production (*Baev et al., 2006*). A similar time-dependent growth inhibition is observed for WT-EmrE in the presence of harmane, and we have previously shown that this substrate triggers uncoupled proton leak through EmrE (*Spreacker et al., 2022*). Thus, while Δ107-EmrE competently performs the proton-coupled drug antiport necessary to confer resistance to toxic substrates, it is detrimental to *E. coli* in the absence of known small molecule substrate in a manner suggestive of proton leak.

To directly test this hypothesis, we measured proton leak in proteoliposomes. The pH-sensitive dye pyranine was encapsulated inside proteoliposomes at pH 6.5, and the liposomes were then diluted 100-fold into pH 7.5 buffer. If protons leak out of the liposome (down the proton concentration gradient), the internal pH will rise and pyranine fluorescence will increase. We compared the

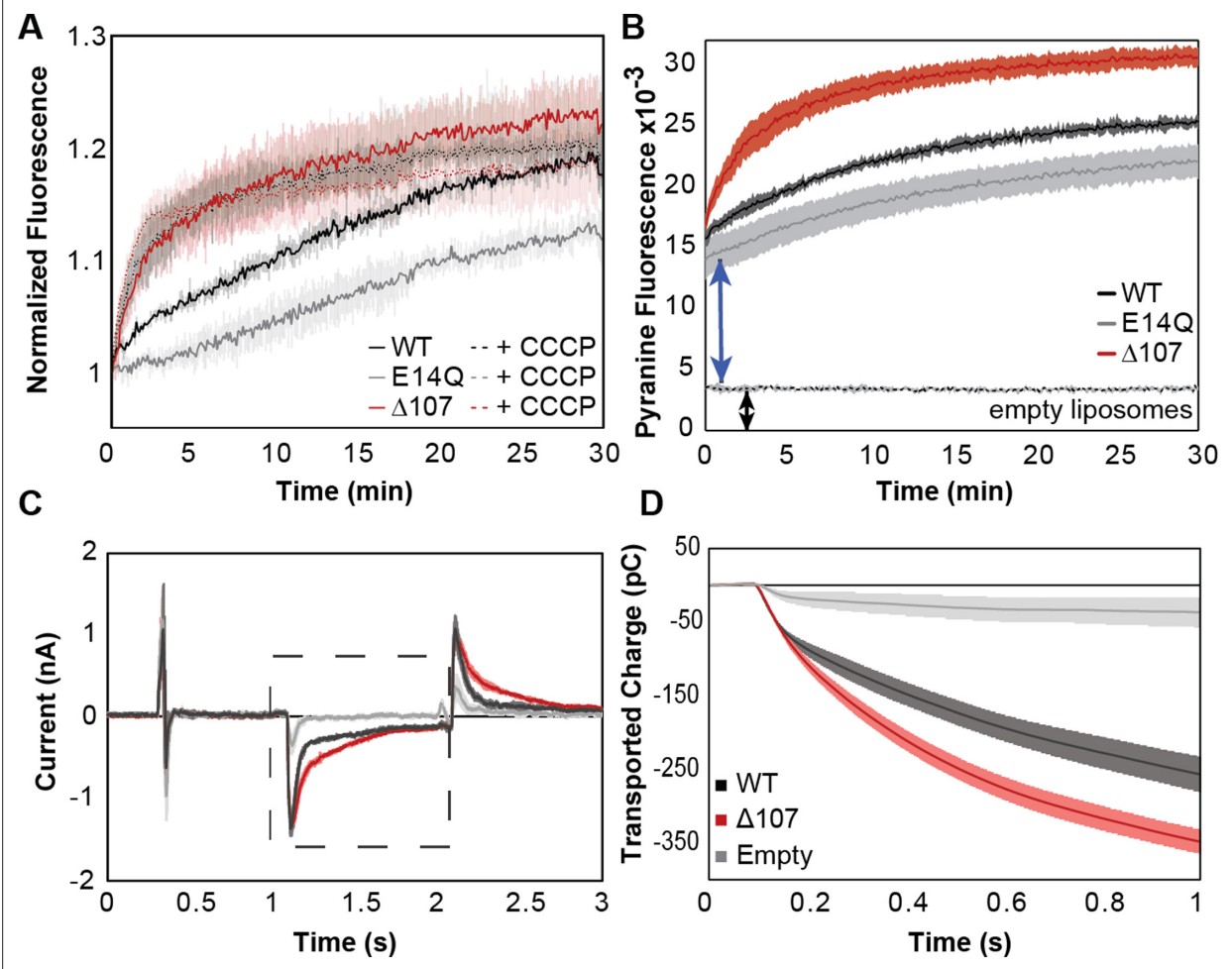

**Figure 3.** C-terminal tail truncation enhances proton leak. (**A, B**) Pyranine fluorescence directly reports on proton leak through EmrE. (**A**) WT (black), Δ107 (red, to distinguish in vitro assays from the cellular assays of *Figure 2*), or E14Q-EmrE (gray) proteoliposomes with 1 mM internal pyranine and internal pH 6.5 were diluted 100-fold into pH 7.5 buffer (solid lines) or pH 7.5 buffer with CCCP (dashed lines) and fluorescence was normalized to time zero. CCCP is a protonophore, providing a positive control for maximal proton leak under these conditions. (**B**) Pyranine fluorescence normalized by subtracting the fluorescence of proteoliposomes diluted into pH 6.5 (no gradient, baseline) from the fluorescence of proteoliposomes diluted into pH 7.5 (transport) shows intraliposomal pH change with proteoliposomes in the lag time prior to initial fluorescence read and increased intraliposomal pH change for Δ107-EmrE than WT-EmrE or E14Q-EmrE. (**C–D**) Solid supported membrane electrophysiology data shows measurable charge movement through WT- and Δ107-EmrE proteoliposomes in the presence of a pH gradient alone, as compared to empty liposomes, with increased charge transport through Δ107-EmrE. (**C**) Current is recorded in real time as a matching pH internal buffer (pH 6.5) is flowed over the liposomes to establish baseline, then a higher pH (pH 7) buffer is flowed over the liposomes to create an outwardly directed proton gradient (dashed box), and finally, the initial buffer (pH 6.5) is flowed back over the liposomes to reverse the charge movement and return to baseline. (**D**) The recorded current during the period of the applied gradient (dashed box, **C**) is integrated to determine the transported charge during that time. In all cases, Δ107-EmrE shows increased proton leak compared to WT-EmrE and controls. The error bars show the standard deviation across three replicates or sensors.

The online version of this article includes the following figure supplement(s) for figure 3:

**Figure supplement 1.** Averaged currents and integrated transport curves of WT-EmrE, Δ107-EmrE, and empty liposomes in the presence of different pH gradients.

**Figure supplement 2.** Averaged currents and integrated transport curves of WT-EmrE and Δ107-EmrE with different lipid to protein ratios (LPRs).

fluorescence, normalized to time zero, of proteoliposomes with WT, Δ107, or E14Q-EmrE. WT-EmrE proteoliposomes show a gradual increase in internal pH over time (*Figure 3A*, solid black), which is faster than the pH change for E14Q-EmrE proteoliposomes (*Figure 3A*, solid gray). This is consistent with a small amount of proton leak through WT-EmrE and a role for E14 in mediating leak. Δ107-EmrE proteoliposomes show a much faster rise in internal pH in this assay (*Figure 3A*, solid red). Repeating the experiment with the protonophore CCCP in the external buffer results in rapid proton leak for all

proteoliposome samples (*Figure 3A*, dotted lines), and the results match the timescale and amplitude of proton leak observed for the Δ107-EmrE proteoliposomes. Thus, C-terminal tail truncation causes rapid proton leak through EmrE.

Initiation of the assay by dilution complicates measurement of the fluorescence baseline, obscuring rapid changes in the few seconds between dilution and initial fluorescence read. We repeated the assay with side-by-side dilution of proteoliposomes (internal pH 6.5) into pH 6.5 buffer (baseline), or pH 7.5 buffer (transport). This baseline normalization reveals a rapid internal pH change (*Figure 3B*). The empty liposome control is flat, showing that the liposomes do not leak without EmrE. However, the signal is non-zero (*Figure 3B*, black arrow), likely due to residual exterior pyranine. The time 0 fluorescence of WT-, E14Q-, and Δ107-EmrE proteoliposomes is much higher (*Figure 3B*, blue arrow), indicating an additional rapid change in the internal pH in the presence of protein. While the encapsulated pyranine is protected from the direct impact of external pH change, proton transport (leak) through EmrE will change internal pH. The transmembrane pH gradient may also affect EmrE itself, altering the pKa of key residues (E14, H110) and causing rapid release of protons inside the liposome, or cause a rapid burst phase of leak as the transporter transitions from a symmetric-pH conformation to an asymmetric-pH conformation.

To further assess the initial rapid pH change, we used solid supported membrane electrophysiology (SSME) to measure ΔpH-driven current in proteoliposomes since this technique provides a continuous readout as a gradient is applied. Many proteoliposomes can be adsorbed onto the gold-coated sensor, enabling highly sensitive detection of electrogenic transport. The same lipid to protein ratio was used as in the pyranine assay, and reported values are an average of three independently prepared sensors per mutant to account for variability in liposome adsorption onto sensors. The liposomes are first equilibrated with external buffer identical to the interior, and then a different external buffer is rapidly washed over the liposomes to create a transmembrane pH gradient while recording is in progress. The capacitive current is measured (*Figure 3C*, *Figure 3—figure supplement 1, left A, B*) and integrated to yield the total transported charge in response to the applied gradients (*Figure 3D*, *Figure 3—figure supplement 1, right*, *Figure 3—figure supplement 2C*). In the absence of drug, empty liposomes have minimal charge movement as expected for minimal proton leak. However, both WT- and Δ107-EmrE have measurable current in the presence of a pH gradient, and Δ107-EmrE has consistently higher leak under all pH conditions (*Figure 3—figure supplement 1*). The consistency of the SSME and pyranine assay results establishes the validity of the assay for comparing WT- and Δ107-EmrE proton leak and the ability to perform multiple assays with the same proteoliposome sensors to compare flux at different absolute pH or gradient magnitudes (*Figure 3—figure supplements 1 and 2*).

Both WT- and Δ107-EmrE also show increased net charge movement at higher absolute pH (*Figure 3—figure supplement 1*). In symmetric pH environments, the only amino acid side chains with pKa values near neutral pH are E14 (pKa 6.8 ± 0.1 and 8.5 ± 0.2 at 25°C) and H110 (6.98 ± 0.01, 7.05 ± 0.02) (*Thomas et al., 2018*; *Morrison et al., 2015*). Δ107-EmrE is lacking Histidine (H110), so any proton binding/release from H110 that contributes to the capacitive current will be absent in Δ107-EmrE, but the net charge transport (*Figure 3D*, *Figure 3—figure supplement 1, right*, *Figure 3—figure supplement 2C, D*) is greater for Δ107-EmrE than for WT-EmrE. This rules out a simple model where proton binding/release from H110 accounts for the fast proton flux. Any conformational change involving the C-terminal tail that contributes to the capacitive current should also decrease as average pH increases and H110 protonation and net charge decrease. However, the opposite pH dependence is observed for WT-EmrE, indicating that this is unlikely to be a major contributor to the SSME current. Furthermore, any rapid release of protons from E14 upon gradient formation will be decreased at high pH as the initial protonation state, and thus the number of protons that can be released, is reduced. Thus, the increased charge movement at higher absolute pH must be due to increased proton leak since there is no other substrate present in these experiments. *Figure 3—figure supplement 2* shows the transient current and net transported charge at low pH, high pH, and under conditions where a drug gradient drives transport. These assays show multiphasic behavior, and close examination of the data with different lipid to protein ratios under each experimental condition further distinguishes pre-steady state (proton release, conformational change) and steady state (transport) processes that contribute to net charge movement. A model where movement of the C-terminal tail regulates access to E14 and C-terminal truncation alters this gating process would be consistent with the observed

currents and their pH dependence (*Figure 3C–E*, *Figure 3—figure supplements 1 and 2*), as well as the longer-timescale change in intraliposome pH (*Figure 3B*). We note that the 1-s SSME traces do not reach equilibrium as the current is not zero and net transported charge is still changing at the end of the assay (*Figure 3D*, *Figure 3—figure supplements 1, right and 2*), but do provide insight into the dead time of the pyranine assay (*Figure 3B*) and match the relative magnitude of the observed burst phase. Altogether, this data supports a role for the C-terminal tail as a secondary gate that minimizes proton leak through WT-EmrE in the absence of substrate.

## The pH-dependent rate of alternating access in Δ107-EmrE is distinct from WT-EmrE

We next used solution NMR to assess the impact of C-terminal tail truncation on the structure and dynamics of EmrE, since this will impact gating and transport. Due to the asymmetric structure of EmrE, the two subunits have unique chemical shifts. As EmrE undergoes alternating access, the two subunits swap conformations, resulting in exchange between AB and BA dimer topology (*Morrison et al., 2015*). The rate of the alternating access exchange process affects the NMR line shape, resulting in distinct sets of peaks for each subunit when exchange is slow, line broadening as the rate increases, and eventually coalescence into a single set of peaks at the average chemical shift when exchange is

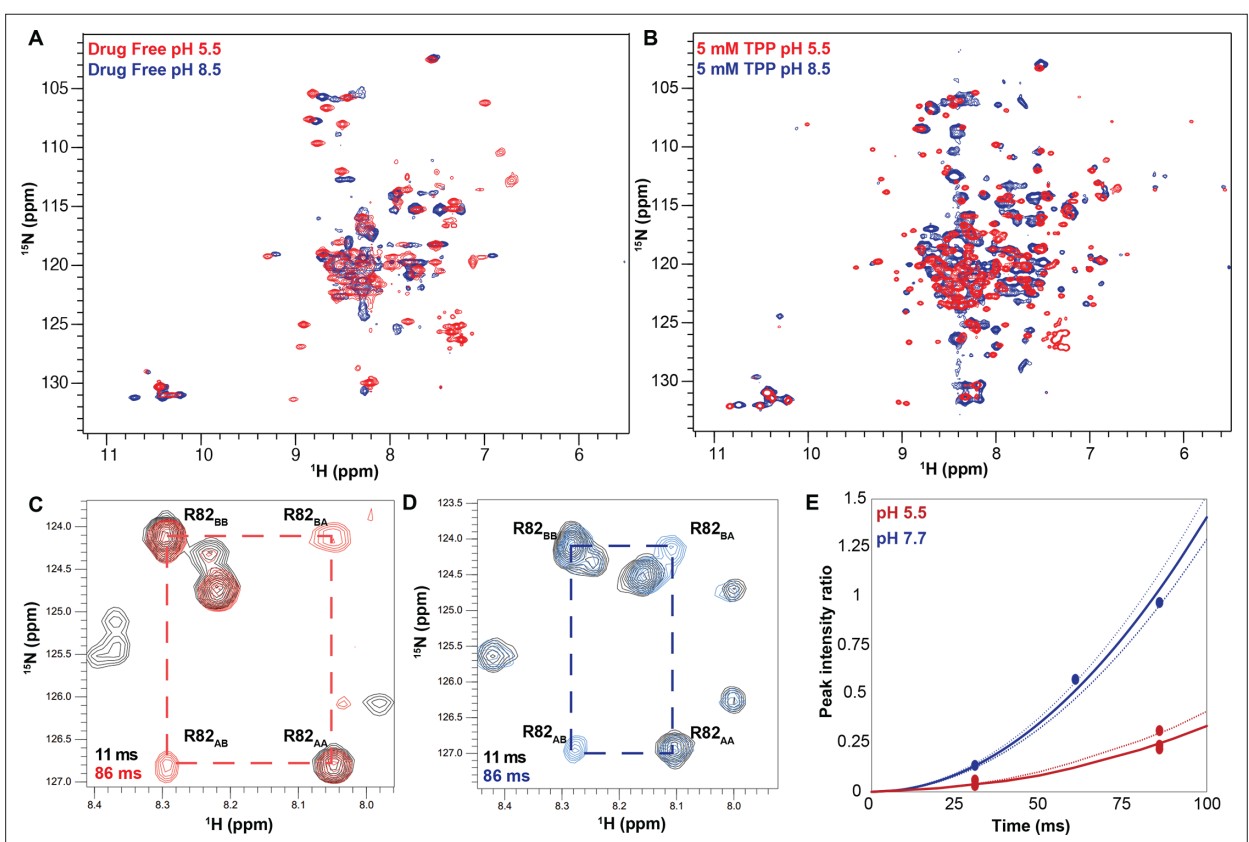

**Figure 4.** The pH dependence of alternating access in Δ107-EmrE is distinct from WT-EmrE. TROSY-HSQC spectra of Δ107-EmrE in the absence (**A**) and presence (**B**) of the tight-binding ligand tetraphenylphosphonium (TPP⁺). While drug binding slows the dynamics of the protein at both low (red) and high (blue) pH, as evident by the better spectral quality in B, in both drug-free and drug-bound Δ107-EmrE, the dynamics of the mutant are highly sensitive to the pH conditions. ZZ-exchange spectroscopy of Δ107-EmrE bound to TPP⁺ was used to quantify the alternating-access rates at low and high pH. ZZ-exchange spectra with the indicated delays are shown for (**C**) pH 5.5 and (**D**) pH 7.7. (**E**) The composite peak intensity ratios for F78, G80, R82, L83, and R106 fit to an exchange rate of $4 \pm 1\ \mathrm{s^{-1}}$ at pH 5.5. At pH 7.7, the composite peak intensity ratios for G80, R82, L83, and R106 fit to an exchange rate of $17 \pm 3\ \mathrm{s^{-1}}$.

The online version of this article includes the following figure supplement(s) for figure 4:

**Figure supplement 1.** Overlay of drug-bound WT- and Δ107-EmrE at low and high pH.

**Figure supplement 2.** Full ZZ-exchange TROSY-HSQC spectra of the low and high pH Δ107-EmrE with TPP⁺.

fast. Thus, the appearance of simple 2D $^1$H-$^{15}$N TROSY-HSQC spectra can provide significant insight into both the structure and dynamics of the transporter under different conditions.

In the absence of drug, Δ107-EmrE is in fast-intermediate exchange at both pH values, with only one set of peaks and significant line broadening (*Figure 4A*). However, the spectra are distinct, with slightly more line broadening at high pH. This indicates that protonation of E14 still affects the overall structure of Δ107-EmrE, and alternating access is slightly slower at high pH. WT-EmrE has similar fast exchange behavior at low pH, but increasing pH results in a significantly slower rate of alternating access (*Figure 4—figure supplement 1*; *Morrison et al., 2015*).

Upon addition of a tight binding drug-substrate, TPP$^+$ to Δ107-EmrE at low pH, two distinct peaks become visible for each residue (*Figure 4B*, red). This confirms that the poor spectral quality of the substrate-free spectrum was due to protein motion and not degradation or aggregation, and that alternating access is significantly slower with TPP$^+$ bound. The similarity of this spectrum of Δ107-EmrE bound to TPP$^+$ at low pH with the spectrum of WT-EmrE under the same conditions (low pH, TPP$^+$ bound) also provides additional evidence that the general structure of Δ107-EmrE remains intact and the binding site has undergone minimal perturbation upon truncation of the last four amino acids (*Figure 4—figure supplement 1A*).

At high pH, TPP$^+$-bound Δ107-EmrE shows significant line broadening and partial coalescence of the two distinct sets of peaks in the NMR spectrum, indicating that the rate of alternating access is faster (*Figure 4B*, blue). We quantitatively measured the rate of alternating access as a function of pH for TPP$^+$-bound Δ107-EmrE with $^1$H-$^{15}$N TROSY-ZZ-exchange NMR experiments (*Li and Palmer, 2009*). In this experiment, a delay is inserted in the pulse sequence between recording the $^{15}$N and $^1$H chemical shifts, such that a conformational exchange during this delay will result in the appearance of cross-peaks with the $^{15}$N chemical shift of the original state and $^1$H chemical shift of the final state (*Figure 4C, D*). By comparing the intensity of these cross-peaks relative to the auto-peaks as a function of the delay time, we can determine the rate of alternating access (*Miloushev and Palmer, 2005*; *Figure 4E*, *Figure 4—figure supplement 2*). The alternating access rate for TPP$^+$-bound Δ107-EmrE is 4 ± 1 s$^{-1}$ at low pH, and 17 ± 3 s$^{-1}$ at high pH. There is greater scatter in the peak intensity ratio at high pH due to enhanced exchange with water for residues on the open face of the transporter, which reduces the peak intensity. However, there is no overlap between low pH and high pH, clearly demonstrating a significant change in alternating access rate for Δ107-EmrE with pH. TPP$^+$-bound WT EmrE has the same rate of alternating access as Δ107-EmrE at low pH, but does not vary significantly with pH (*Morrison and Henzler-Wildman, 2014*). Thus, truncation of the C-terminal tail alters the pH dependence of alternating access for Δ107-EmrE in both the absence and presence of drug substrates, supporting a role for this region in regulating the pH-dependent conformational dynamics of EmrE.

## The C-terminus controls a water wire into the primary binding site

To further investigate how the C-terminal tail of EmrE may interact with other regions of EmrE and gate access into the transport pore, we carried out MD simulations of substrate-free WT-EmrE and Δ107-EmrE in a DMPC lipid bilayer. The protonation states were set to simulate a pH between 7.0 and 8.0, where only E14$^A$ with the higher pKa is protonated. We used an NMR structure determined with TPP$^+$ (PDB: 7JK8) as the initial structural model (*Shcherbakov et al., 2021*). Since this NMR structure does not include the C-terminal tail, we modeled it with CHARMM-GUI (*Lee et al., 2016*). The systems were first equilibrated at constant temperature 310 K and 1 bar pressure for 400 ns, while position restraints were gradually released. After this equilibration, the RMSD of the protein compared with the initial structure plateaued. The production simulations were run for another 1000 ns, and MD trajectories were output every 0.1 ns. With the same starting structure, we ran three parallel replicas to ensure consistency (*Figure 5*, *Figure 5—figure supplements 1 and 2*).

A critical prerequisite of proton transport is a water wire, either transient or long-lasting, that allows protons to transport through the Grotthuss hopping mechanism (*Agmon, 1995*). In the Δ107-EmrE MD simulations, we identified a water chain (aka 'water wire') not seen in WT simulations that connects E14 at the primary binding site to bulk water. It enters the protein from the open side near R106$^A$, passing through the triad of A61$^A$, I68$^B$, and I71$^B$, and then comes into the primary binding site (*Figure 5C*). To quantitatively understand the connectivity of this water wire over time, we calculated the length for the shortest water path $S$ for each frame in the trajectory with graph theory. This

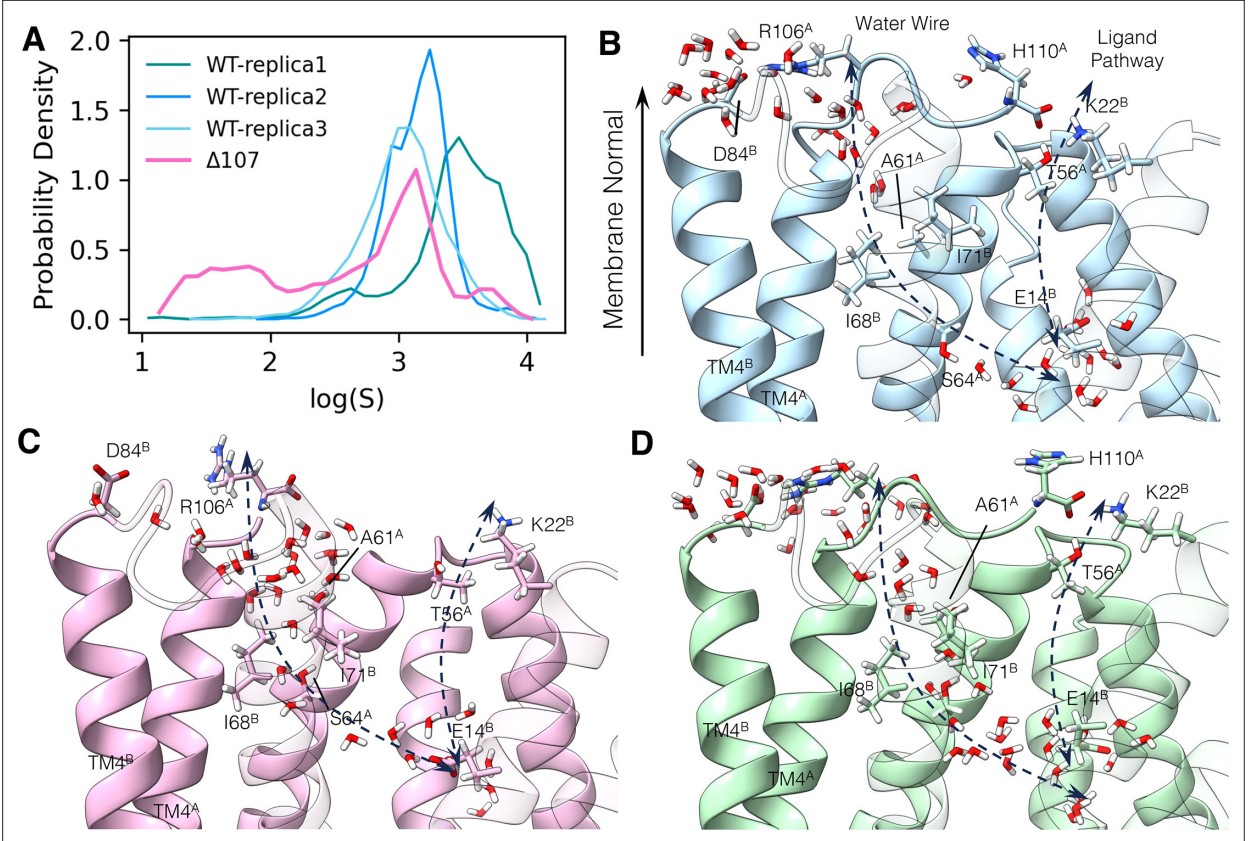

**Figure 5.** The C-terminus tail caps the water wire from the open side. (**A**) Logarithm of the minimum water distance log($S$) histogram. (**B–D**) The following panels illustrate a few snapshots in the simulation. The membrane normal vector points to the open side of EmrE. Two dashed arrows show the ligand pathway and the water chain, respectively. TM1 to TM3 in subunit B is shown transparently to better illustrate the interface between the two subunits. (**B**) Dry snapshot of WT-EmrE. (**C**) Wet snapshot of Δ107-EmrE. (**D**) A rare event snapshot when WT-EmrE is hydrated. The color codes are the same as in **Figure 6**. Yellow stars highlight the backbone of the C-terminal residue (R106 or H110), and the yellow arrowhead (**B, D**) highlights the backbone of R106 in the full-length construct to illustrate where the tail would terminate in Δ107.

The online version of this article includes the following figure supplement(s) for figure 5:

**Figure supplement 1.** Autocorrelation function of dihedral angles of the tail.

**Figure supplement 2.** Cross-correlation function of dihedral angles of the tail at four given lag times.

**Figure supplement 3.** Overlay of initial and final structures from three replicates.

**Figure supplement 4.** Hydrogen bonding of D84 and the C-terminal tail.

length is defined such that smaller values for $S$ reflect better connectivity of the water molecules, as described in more detail in the methods section (*Li and Voth, 2021a*). The logarithm of the shortest path, log($S$), is plotted for all simulation systems (*Figure 5A*). For Δ107-EmrE, there is a leak state characterized by a much smaller log($S$) where the water wire is very well connected, while for WT-EmrE, log($S$) is consistently large. The existence of this water chain in Δ107-EmrE is consistent with the enhanced proton leak observed experimentally. This newly found water wire starts very close to the C-terminus and is distinct from the ligand entry path (*Jurasz et al., 2021*).

## Structural basis for the C-terminus gating

In the initial structure of WT-EmrE, the C-terminus is floating in bulk water. After equilibration, we observed the tail coming closer and interacting with the protein in all three replicas (*Figure 5—figure supplement 3*). The tail has two notable interactions with other parts of the protein, the first of which is a salt bridge between D84 and R106. The second involves the carbonyl group of the C-terminus, which forms hydrogen bonds with T56 and occasionally forms a salt bridge with K22 (*Figure 5—figure supplement 4*). When these interactions occur, the tail moves near the water chain. Examining

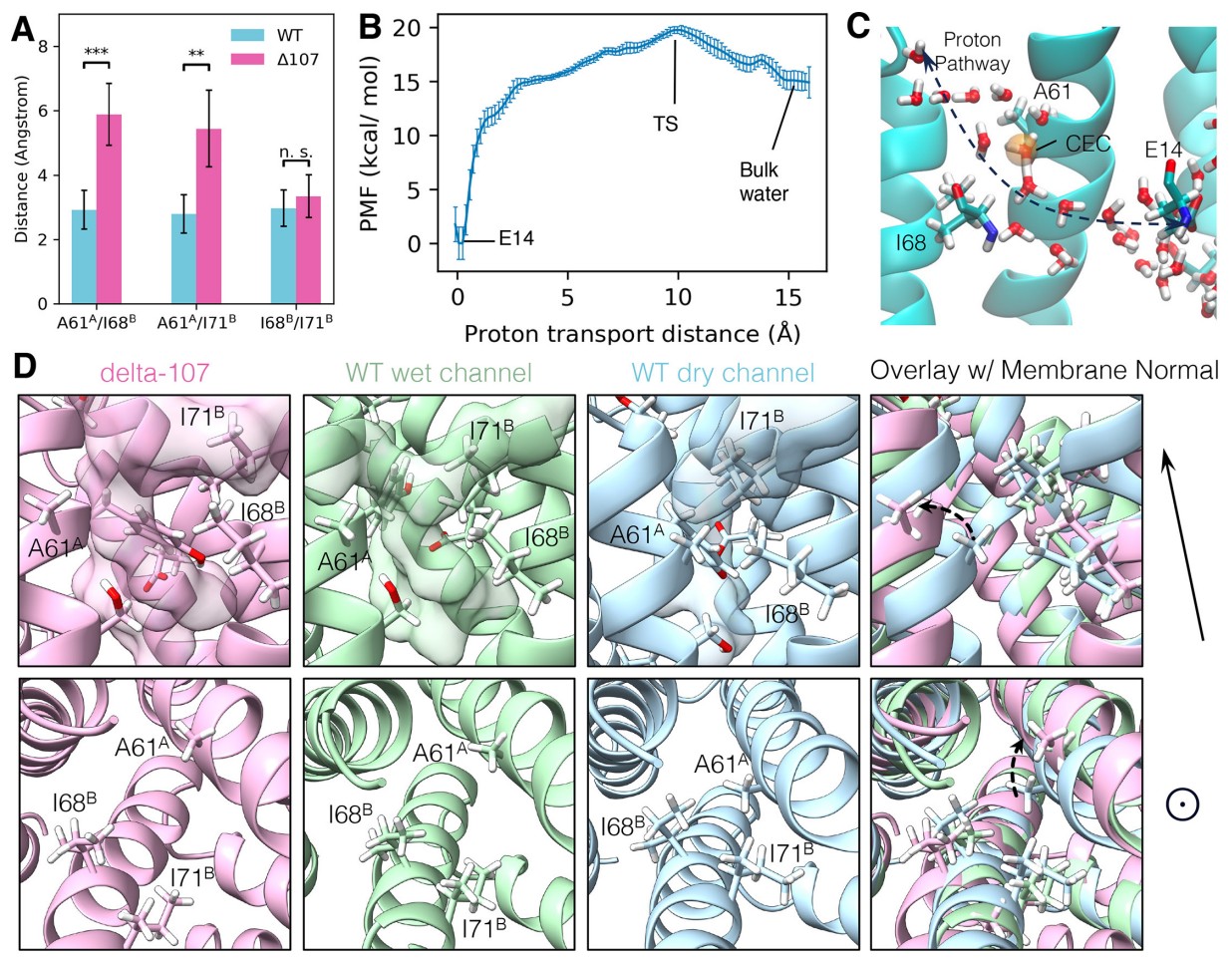

**Figure 6.** The structural basis of C-terminal gating. (**A**) The minimum distance between side chain hydrogens for A61[A], I68[B], and I71[B]. In WT-EmrE, the side chain of A61[A] is significantly closer to I68[B] and I71[B], while the distance between I68[B] and I71[B] does not change significantly. The error bars show the standard deviation along the trajectory. All p-values were calculated from a two-sided *t*-test, **p < 0.01, ***p < 0.001. (**B**) The proton transport potential of mean force (PMF), as a function of the distance between the center of the excess charge (CEC) and the donor (E14) on the direction of transport (see *Equation 2* in Methods). Error bars show standard deviation from a block analysis of 5 blocks. (**C**) A snapshot of the transition state. The orange sphere is the proton CEC. (**D**) Conformations of the A61[A], I68[B], I71[B] triad from two different angles. The membrane normal shown at the right points to the open side of EmrE. The upper panels are from a side view, and the lower panels are looking top–down into the primary binding site from the open side. The transparent surface in the upper panels shows the water wire. Unlabeled residues shown as stick representation are E14[B], Y60[A], and S64[A].

The online version of this article includes the following figure supplement(s) for figure 6:

**Figure supplement 1.** Average Cartesian coordinates of umbrella windows.

characteristic snapshots from the simulation trajectories illustrates the effect of the C-terminal tail on water wire formation. *Figure 5B* shows the worst-hydrated snapshot of WT-EmrE determined by the highest *S*. *Figure 5C* shows the best-hydrated snapshot of Δ107-EmrE determined by the lowest *S*. Despite the overall dryness in the channel of WT-EmrE, there were a few rare moments where a transient water wire formed, characterized by a sudden drop in water path length, and *Figure 5D* shows the best-hydrated snapshot from that simulation. In WT-EmrE, the water wire is broken at the triad of A61[A], I68[B], and I71[B] (*Figures 5B and 6D*), suggesting that these three hydrophobic residues may act as a bottleneck for the water wire.

To test this hypothesis with statistically meaningful results, we calculated the distances between the closest pairs of side-chain hydrogens among these three residues for the whole trajectory. For WT-EmrE, the minimum hydrogen–hydrogen distances of A61[A]–I68[B] and A61[A]–I71[B] are 2.9 ± 0.6 and 2.8 ± 0.6 Å, respectively, and for Δ107-EmrE, these distances increased to 5.9 ± 1.0 and 5.4 ± 1.2 Å (*Figure 6A*). The I68[B]–I71[B] distance does not differ significantly. These increased distances suggest

that A61$^A$ is moving away in the Δ107 variant, opening a pore for the water wire to form. For comparison, the diameter of a water molecule is measured to be around 2.7 Å (*Schatzberg, 1967*). This means a triangle larger than 5.4 Å in size is required for a water molecule to fit inside. Additionally, the most hydrated snapshot in WT-EmrE showed A61$^A$ takes a conformation more similar to Δ107-EmrE and very different from dry WT-EmrE. This confirms the role of A61$^A$ rotation in controlling the water wire formation.

Experimental testing of this hypothesis by mutagenesis is complicated by the small size and anti-parallel topology of EmrE, as many residues play multiple functional roles, and mutation of any of these residues will perturb not only the proposed hydrophobic gate (A61$^A$, I68$^B$, and I71$^B$) but also the close packing necessary to close the transporter on the opposite face of the membrane where A61$^B$, I68$^A$, and I71$^A$ are located. Prior scanning mutagenesis replacing A61, I68, and I71 with alanine, valine, glycine, or cysteine (*Amadi et al., 2010*; *Mordoch et al., 1999*; *Wu et al., 2019*) did not impact the ability to confer resistance to common EmrE substrates, such as ethidium, acriflavine, or methyl viologen. However, the mutation to tryptophan severely impaired ethidium resistance (*Lloris-Garcerá et al., 2013*), and mutation of any of these residues to cysteine impacts substrate binding (*Amadi et al., 2010*), demonstrating that these residues are functionally important.

## The potential of mean force of proton transport supports a hydrophobic bottleneck

The potential of mean force (PMF) for explicit proton transport (see Methods) shows the free energy change of the system as a function of a particular collective variable (CV) or reaction coordinate. It can provide additional information beyond structural snapshots of a reaction, which are only incomplete samples of the ensemble. For example, the existence of a water chain does not necessarily mean good proton conductance, but by contrast, the PMF for explicit proton transport (including Grotthuss proton shuttling) can provide key thermodynamic and kinetic information about the transport process (*Li and Voth, 2021a*; *Ilan et al., 2004*). However, free energy sampling involving proton transport is also intrinsically complicated. First, proton transport involves chemical bond breaking and formation, which is beyond the capability of classical MD, so expensive quantum mechanics/molecular mechanics may be required. Second, when an excess proton is solvated in the water, the net positive excess charge defect arising from the presence of the excess proton can be delocalized in the water network. It is not possible to define which proton is exactly the 'excess proton' as Grotthuss shuttling dynamically rearranges these definitions. To address these issues, we have developed a method called Multiscale Reactive MD (MS-RMD) (*Kaiser et al., 2024*). It can model bond forming and breaking involving excess proton shuttling at a computational cost near classical MD. In this approach, one can also conveniently define a 'center of excess charge' (CEC) to describe the location of the excess positive charge defect.

We carried out umbrella sampling with MS-RMD that describes the proton transport from E14$^B$ to the bulk water in WT-EmrE. The CV '$x$' is defined similarly as in reference (*Li and Voth, 2021b*) as the distance between the glutamate oxygen to the CEC, mapped along a vector that aligns with the proton transport direction (see Methods). The resulting PMF (*Figure 6B*, *Figure 6—figure supplement 1*) indicates a deep well near $x = 0$ Å, where the proton is on the glutamate, and a transition state near $x = 10.0$ Å. A conformational snapshot from the transition state shows the CEC (*Figure 6C*, orange sphere) is in close proximity to the pore defined by I68 and A61 (*Figure 6D*). The validity of this PMF calculation can be further supported by the pKa calculation from this PMF. The resulting pKa for this E14 is 7.1, close to the experimental value of 6.8 ± 0.1 at 25°C (or 7.0 ± 0.1 at 45°C) (*Morrison et al., 2015*). This supports the hypothesis that the bottleneck of proton transport is this hydrophobic gate.

## Re-assessing protonation state by NMR

Prior NMR pH titrations of WT-EmrE in the absence of drug-substrate revealed that the two essential E14 residues in the asymmetric homodimer have distinct pKa values (*Morrison et al., 2015*), reflecting their unique structural environments. Upon binding TPP$^+$, one of the E14 residues is protected from protonation and no longer titrates, while the other (E14$^A$) retains a pKa of 6.8 ± 0.1, similar to the drug-free state (*Morrison et al., 2015*). The only other titratable residue previously identified in WT-EmrE is the C-terminal histidine (*Thomas et al., 2018*), which also has a pKa near neutral pH in

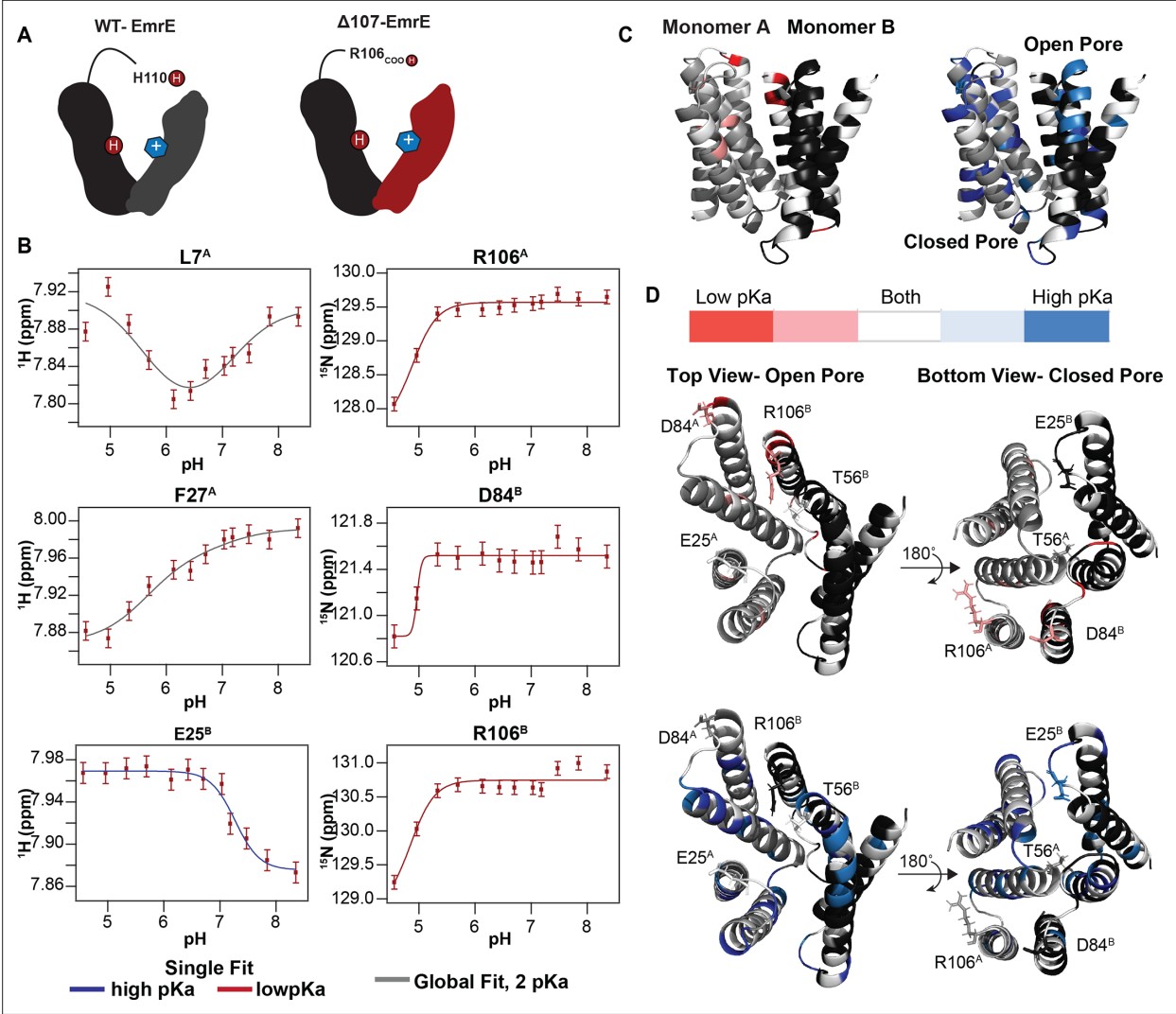

**Figure 7.** pH titration of TPP⁺-bound Δ107-EmrE supports the possibility of secondary gating. (**A**) In WT-EmrE bound to TPP⁺, one E14 residue and one H110 residue are the only titratable sites (dark red circles labeled H⁺). In Δ107-EmrE, H110 is not present, suggesting that only one titratable group should remain (E14). (**B**) The proton and nitrogen chemical shifts (error bars reflect spectral resolution) for individual residues of TPP⁺-bound Δ107-EmrE were recorded as a function of pH. The resulting titration profiles do not show the expected single-pKa pattern. Some are curved, consistent with multiple pKa values, and others are consistent with a single pKa but at either high or low pH. All of the data can be globally fit to two pKa values, using either a 2-pKa fit (5.6 and 7.1, gray) or single pKa fit at the relevant value (5.6, red; 7.1, blue). (**C**) Residues sensitive to each pKa value are plotted on the faRM model (*Vermaas et al., 2018*) using the indicated color scale. (**D**) Residues strongly sensing the lower pKa value cluster around the C-terminus (R106) and 3–4 loop (includes residue D84) on both the open and closed face of the transporter, while the 1–2 loop (includes residue E25) and T56 on the open side of the pore sense both pKa values (left).

The online version of this article includes the following figure supplement(s) for figure 7:

**Figure supplement 1.** NMR pH titration of TPP⁺-bound Δ107-EmrE.

WT-EmrE. Since H110 was removed by truncation in Δ107-EmrE, we expected only one protonation event in NMR pH titrations of TPP⁺-bound Δ107-EmrE (*Figure 7A*), corresponding to the one E14 residue that remains titratable when substrate is bound. A single protonation event would normally result in linear change in peak position from the chemical shift of the protonated state to the chemical shift of the deprotonated state over the course of the titration. This is because proton on-/off- is almost always in the fast-exchange limit for NMR, resulting in the observation of a single peak at the population-weighted average chemical shift at each titration point. However, several peaks exhibit distinctly curved titration paths with transitions in two different pH ranges (*Figure 7—figure supplement 1*) for TPP⁺-bound Δ107-EmrE. Plotting the chemical shift of well-resolved peaks as a function of

pH yields titration curves that can be fit using standard pKa equations. In this case, the data is well fit with a global 2 pKa model yielding apparent pKa values of 5.6 ± 0.2 and 7.1 ± 0.2. The higher of these two pKa values is close to the E14$^A$ pKa in TPP$^+$-bound WT-EmrE, and the residues most sensitive to this protonation event are found lining the transport pore near E14$^A$, supporting the assignment of this pKa to E14$^A$ (*Figure 7D*).

This leaves the lower pKa unaccounted for. Possibilities include other acidic residues in the loops, E25 or D84, or the C-terminal carboxylate itself (now at R106 in the Δ107-EmrE construct). The residues most sensitive to this lower pKa include R106 and the tail of subunit A, the TM1–TM2 loop of subunit B, and TM3–TM4 loop of monomer B, all of which are on the same 'open' face of EmrE (*Figure 7C*). Residues E25 and D84 are located in these loops and have previously been suggested to be part of a secondary binding site for substrates like TPP$^+$ (*Glaubitz et al., 2000*). There is no evidence that these residues titrate in this pH range in WT-EmrE or in other mutants for which we have carried out NMR pH titrations (*Robinson et al., 2017*; *Morrison et al., 2015*; *Wu et al., 2019*), making it unlikely that E25 or D84 titrate in this pH range in the full-length transporter. However, truncation of the C-terminus in Δ107-EmrE could alter the structure, environment, and pKa of these residues. Indeed, the hydrogen bond between R106 and D84 observed in the WT-EmrE simulations (*Figure 5B, D*) is broken when the tail is truncated in Δ107-EmrE (*Figure 5C*), and D84 and R106 have some of the largest pH-dependent chemical shift changes. There are relatively few experimental reports of the pKa of the terminal carboxylate in proteins, but it has been reported to have a pKa as high as 5.9 for the partially buried C-terminus of subunit *c* of $F_1F_0$ ATP synthase (*Grimsley et al., 2009*). MD simulations show the C-terminal carboxylate in WT-EmrE hydrogen bonds with T56 (TM 2–3 loop) and K22 (TM 1–2 loop) on the open face of the transporter (*Figure 5B, D*), but the C-terminus in Δ107-EmrE no longer interacts with these residues (*Figure 5C*). Examining the residues that sense this lower pKa shows that the TM3–TM4 and TM1–TM2 loops on the open face have larger chemical shift changes associated with the low pKa protonation event than those loops on the closed face of EmrE, while the C-terminal tail residues in both subunits detect the lower pKa. Comparison with the MD simulations shows that the residues involved in the hydrogen bond networks anchoring the C-terminal tail over the pore align well with the full list of residues that are sensitive to the lower pKa in the NMR titrations, including T56$_A$ (TM2–TM3 loop on the 'open' face), V15–G17, I37, Y40, V69, and S72–L73. Thus, although this lower pKa is likely an artifact of tail truncation, this data experimentally supports the importance of the interactions between the tail and the rest of EmrE identified in the MD simulations as important for disrupting the water wire and occluding the E14-binding site.

## Proton leak through Δ107-EmrE does not synergize with harmane

The well-established function of EmrE is proton-coupled antiport of toxic substrates, leading to toxin efflux and drug resistance. Recently, we discovered that some substrates, such as harmane, instead trigger uncoupled proton uniport, leading to ΔpH dissipation and defects in NADH production and growth in *E. coli* (*Spreacker et al., 2022*), essentially causing susceptibility rather than resistance. We suspected the enhanced proton leak observed through Δ107-EmrE and this harmane-triggered proton leak might have common elements in their underlying mechanism. Using SSME, we first compared the inherent proton leak through WT- and Δ107-EmrE in the absence of substrate. For the same ΔpH driving force, Δ107-EmrE has ≈3-fold greater proton leak than WT-EmrE (*Figure 8A, D*). However, upon addition of 16 μM harmane, there is a large increase in proton leak through WT-EmrE and a small increase in leak through Δ107-EmrE, such that this substrate triggers identical total leak through either transporter (*Figure 8B, D*, *Figure 8—figure supplements 1 and 2*). This SSME-detected proton leak increases with harmane concentration and is saturable in both WT- and Δ107-EmrE (*Figure 8C*, *Figure 8—figure supplements 1 and 2*). If harmane acts as an allosteric regulator of the transporter that can unlock the secondary gate, then a saturating amount of harmane will result in the maximal signal for the WT transporter as observed. This was also confirmed in a pH-detected liposomal assay where addition of harmane decreases the magnitude of the pH change upon addition of CCCP to WT-EmrE containing proteoliposomes relative to empty liposomes (*Figure 8—figure supplement 3*). In Δ107-EmrE, C-terminal truncation removes the majority of the secondary gate and key residues in the allosteric site, rendering proton leak comparably independent to harmane (*Figure 8*, *Figure 8— figure supplements 1 and 2*).

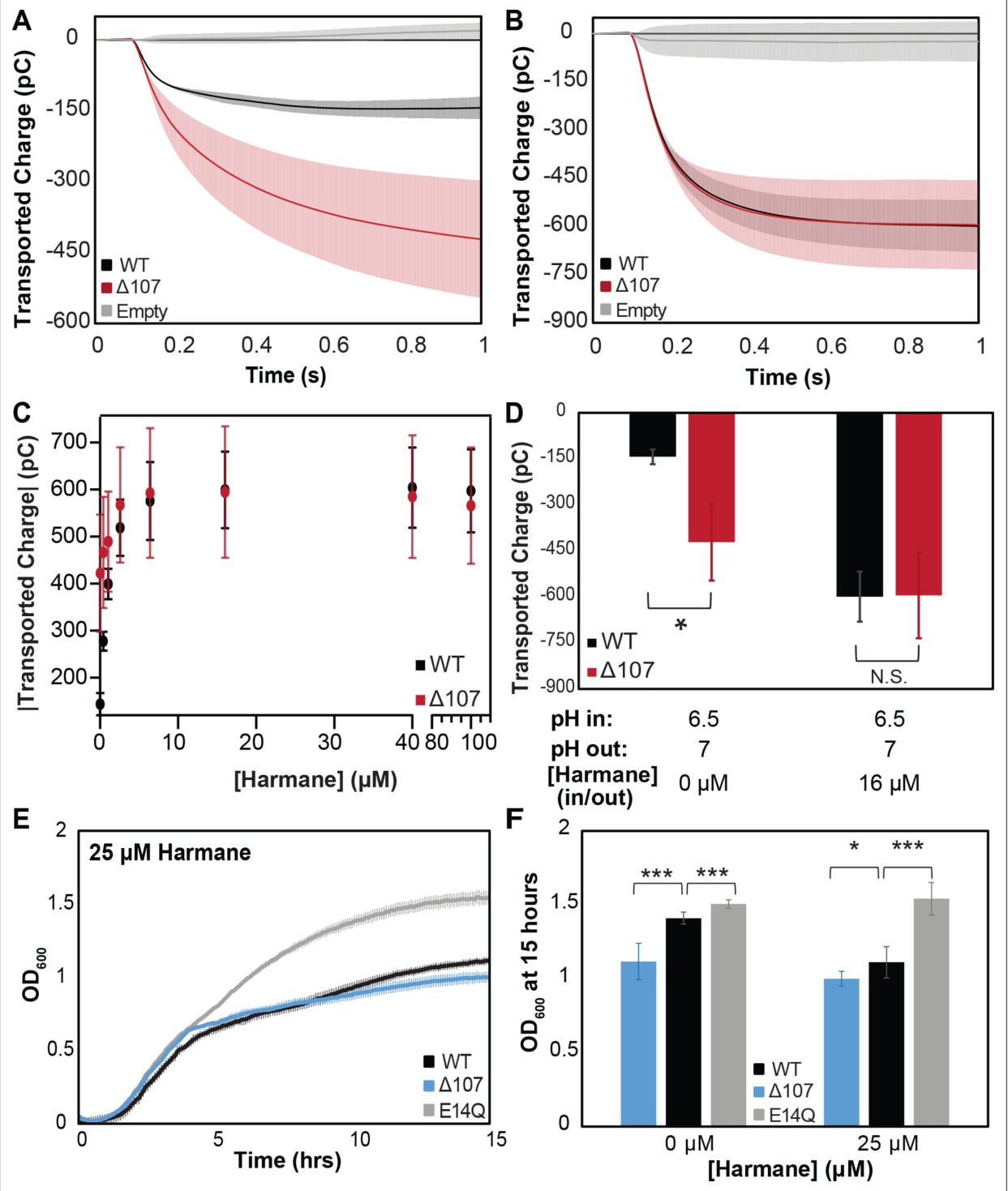

**Figure 8.** Intrinsic leak in Δ107-EmrE does not synergize with harmane-induced leak. Solid supported membrane electrophysiology (SSME) traces of transported charge corresponding to proton leak in the absence (**A**) and presence (**B**) of harmane show that 16 μm harmane induces leak in WT-EmrE that is comparable to the leak observed through Δ107-EmrE in the absence of harmane. In the presence of increasing concentrations of harmane (**C**) the leak signal for WT-EmrE quickly converges to that of Δ107-EmrE. The leak observed for Δ107-EmrE is more variable, displaying larger standard deviations than WT-EmrE proteoliposomes (**C**) This could be due to greater variability in the unregulated transport activity of Δ107-EmrE compared to harmane-gated leak in WT-EmrE, and the impact of this unregulated behavior on the sensitivity of SSME to variation in the absolute number of proteoliposomes adsorbed on the surface sensor. (D) Bar graph of uncoupled proton leak through WT- and Δ107-EmrE proteoliposomes is significant (*) in the absence of drug, but these differences are abolished upon additon addition of 16 μM harmane. Growth assays in the absence of substrate

*Figure 8 continued on next page*

*Figure 8 continued*

(*Figure 1A*) show a clear growth defect for *E. coli* expressing Δ107-EmrE compared to WT-EmrE, which is nearly eliminated when cells are grown in the presence of 25 µM harmane (**E, F**). Δ107-EmrE data is shown in blue for cellular assays and red for in vitro assays to readily distinguish the assay type. The error bars show the standard deviation across three sensors for SSME or across six replicates for growth assays (two biological replicates with three technical replicates each). All p-values were calculated from a two-sided *t*-test, *p < 0.05, **p < 0.01, ***p < 0.001.

The online version of this article includes the following figure supplement(s) for figure 8:

**Figure supplement 1.** Averaged currents of WT-EmrE, Δ107-EmrE, and empty liposomes in the presence of different concentrations of harmane.

**Figure supplement 2.** Integrated transport curves of WT-EmrE, Δ107-EmrE, and empty liposomes in the presence of different concentrations of harmane.

**Figure supplement 3.** pH detected liposomal leak assay shows harmane dissipates ΔpH in an EmrE-dependent manner.

**Figure supplement 4.** Growth assays show a differential impact of harmane on the growth of *E. coli* expressing WT-, E14Q-, or Δ107-EmrE.

To test this theory in the native organism, we conducted in vivo growth assays with WT- and Δ107-EmrE in the presence of harmane. MG1655-ΔemrE *E. coli* cells constitutively expressing WT-, E14Q-, and Δ107-EmrE from a plasmid were grown in the presence of 25 µM harmane. The cells expressing E14Q grew equally well in the presence or absence of harmane, as the mutation of the primary binding site prevents proton binding in the transport pore and abolishes any proton leak (*Figure 8E*). In the presence of harmane, the difference in growth between Δ107- and WT-EmrE is eliminated, with significant growth defect for both constructs relative to E14Q-EmrE (*Figure 8E, F*, *Figure 8—figure supplement 4*). This in vivo data exactly matches the in vitro SSME and pyranine transport assays, demonstrating the importance of the C-terminal tail in gating and allosteric regulation of EmrE in vivo.

## Discussion

There is a growing appreciation that uniport, symport, and antiport simply represent extremes of a unified transport model that includes all possible binding and conformational states and their transitions (*Beckstein and Naughton, 2022*). Despite the expectation that EmrE would have a simple mechanism and clearly illuminate the minimal requirements for coupled transport (*Schuldiner, 2009*), it has proven to be surprisingly complex, exposing unexpected features of membrane protein topology and transport mechanism. The free exchange model, an extension of a universal 8-state transport model to include the ability of EmrE to bind two protons at the two E14 residues in the core of the homodimer (*Robinson et al., 2017*), accounts for most of the available data. It includes all states and transitions observed by NMR and can account for the ability of EmrE to confer resistance to some substrates and susceptibility to other substrates (*Spreacker et al., 2022*). However, this model predicts rapid proton leak through WT-EmrE, while experimental data shows a small proton leak of smaller magnitude and similar timescale to coupled transport. In combination with prior data noting the importance of the C-terminal tail (*Thomas et al., 2018*), the experimental data and MD simulations presented here support a regulatory role of the C-terminal tail as part of a secondary gate that minimizes proton leak in the absence of substrate and can be opened by binding of a drug-substrate.

Prior NMR data (*Spreacker et al., 2022*; *Thomas et al., 2018*; *Glaubitz et al., 2000*) led to the hypothesis that the C-terminal tail acts as a secondary gate occluding the primary E14-defined binding site in the absence of drug-substrate, with drug binding to a peripheral site opening this gate and allowing release of protons from E14. This model can explain the observed coupling of the C-terminal tail with both drug-binding and protonation events at the primary site (*Thomas et al., 2018*) and the correspondence of proton off-rate and substrate on-rate in prior stopped-flow studies of EmrE (*Adam et al., 2007*). Here, we combine MD simulations with experimental studies of a tail-truncated mutant, Δ107-EmrE, to more directly test the tail-gating hypothesis and determine whether this model can explain the minimal proton leak observed for WT-EmrE (*Robinson et al., 2017*) and the newly discovered harmane-gated proton uniport activity of EmrE (*Spreacker et al., 2022*).

If the tail is important for gating proton access to the binding pocket and preventing proton leak through the WT transporter, then truncation should enhance proton leak through EmrE. This is exactly what we observe, with increased uncoupled proton flux through Δ107-EmrE in vitro (*Figure 3*) and diminished growth of *E. coli* expressing Δ107-EmrE in vivo (*Figure 2*). Comparing MD simulations of WT and Δ107-EmrE shows that the C-terminus can interact with TM3 to block formation of a water

wire, providing a structural hypothesis for how the tail gates access to the primary EmrE-binding site at E14 and regulates proton entry and exit from that site, as required for proton leak. Truncation of the C-terminal tail in Δ107-EmrE also removes key residues that are part of a secondary substrate-binding site, reduces the sensitivity to harmane-triggered proton leak in vitro and in vivo (*Figure 8*). Identical maximal harmane-triggered proton leak through WT- and Δ107-EmrE further supports the model that substrates bind at a secondary site in the vicinity of the C-terminal tail and releasing this secondary gate to allow proton flux.

The residues identified as important for regulating the formation of the water wire, A61, I68, and I71, are all highly conserved. An analysis of 369 EmrE-related SMR sequences (*Brill et al., 2015*) shows A61 is fully conserved, while I68 and I71 are highly conserved with valine as the only substitution. D84 is the only fully conserved charged residue other than E14. A more recent analysis of SMR genes within the Joint Genome Institute's Genomic Encyclopedia of Bacteria and Archaea shows A61 and K22 are highly conserved across the SMR family, while I68, I71, T56, and D84 are conserved within the Qac subfamily (*Burata et al., 2022*). A61 and K22 are nearly as well conserved as the GXG motif in TM3 known to act as a fulcrum for conformational exchange between open-in and open-out conformations or the G97 in TM4 that is important for dimerization. A61C is not reactive with NEM (I68 and I71 not tested) (*Mordoch et al., 1999*), consistent with the closed hydrophobic gate observed in the MD simulations. Although drug binding and transport do not report on hydrophobic gating as directly, A61C has impaired resistance to acriflavine and methyl viologen (*Mordoch et al., 1999*), while A61L has impaired growth on ethidium (*Wu et al., 2019*). I68W, I68C, and I71W impair growth on ethidium; I68A and I71G impair resistance to methyl viologen; and I61C, I68W, I68C, I71W, and I71C have impaired TPP$^+$ binding (*Amadi et al., 2010*; *Wu et al., 2019*; *Lloris-Garcerá et al., 2013*). In addition, K22C, T56C, and D84C reduce TPP$^+$ binding *Amadi et al., 2010*; K22C reduces ethidium resistance *Amadi et al., 2010*; and D84C shows reduced resistance to ethidium and methyl viologen (*Yerushalmi and Schuldiner, 2000b*). In methyl viologen uptake assays, substitution of like charge at E25D and R82K resulted in transport comparable to WT, while K22R, D84E, and R106K had impaired uptake indicating a more specific requirement for these positions (*Yerushalmi and Schuldiner, 2000b*). Chemical shift perturbations upon harmane binding also highlight D84 and R106 (*Spreacker et al., 2022*). The secondary gating model and MD simulations presented here provide a rationale for the functional significance of these residues observed in the prior work.

Active transport requires that a transporter is only ever open to one side of the membrane. This is generally thought to require the formation of an occluded state where the substrate-binding site is closed off from both sides of the membrane as the transporter transitions from the conformation open to one side of the membrane to the conformation open to the other side in order to avoid even transient formation of a channel. Often a single gate is thought to control access to the transport pore, but sometimes multiple gates regulate a more complex transport cycle (*Diallinas, 2014*; *Rudnick, 2011*). This is clearly seen in elevator mechanism transporters such as Glt$_{Ph}$. A mobile core domain contains the substrate-binding site and moves up and down relative to the more rigid scaffold domain, effectively transitioning between inward- and outward-occluded conformations. From either of these endpoint occluded states, a small hairpin domain can open to expose the binding pocket for substrate entry or exit. This hairpin gate must close to allow the sliding elevator movement and subsequent gate opening on the other side of the membrane (*Reyes et al., 2009*). Studies of Glt$_{Ph}$ have highlighted the evolutionary benefit of a kinetically controlled transport mechanism and the role of allosteric regulation in opening and closing the gate (*Riederer and Valiyaveetil, 2019*; *Oh and Boudker, 2018*). UapA, the xanthine-uric acid/H$^+$ symporter from the Nucleobase-ascorbate transporter (NAT) family, operates through a similar elevator mechanism, and residues outside of the primary binding site have also been shown to regulate substrate affinity, specificity, and transport dynamics, supporting a role for allosteric regulation of the transport cycle (*Kosti et al., 2010*; *Vlanti et al., 2006*; *Koukaki et al., 2005*; *Papageorgiou et al., 2008*; *Diallinas, 2013*). In the Major Facilitator Superfamily (MFS) sugar transporters, multiple occluded state structures have been identified, suggesting that multiple gates may regulate the function of these transporters as well and may explain the ability of some transporters in this family to switch between proton-coupled sugar symport and uncoupled proton uniport (*Madej et al., 2014*). Here, we show that even very small transporters, such as EmrE, can have complex mechanisms of gating and transport regulation. Within the SMR family, the QAC transporters, including EmrE, are promiscuous transporters with

≈110 amino acids and a highly conserved C-terminal histidine, while the Gdx transporters are selective for guanidinium and are missing the C-terminal tail with a total length of ≈105 amino acids and no C-terminal histidine. Thus, the tail-coupling mechanism may be important for maintaining proton-coupled antiport while transporting a broader array of substrates, but this hypothesis requires further investigation. However, the existence of a secondary gate in EmrE is broadly relevant as phylogenetic analysis has suggested that SMRs may have been the progenitors of the MFS, Bacterial/Archaeal transporters, and drug-metabolite transporter superfamilies as a whole (*Bay and Turner, 2009*; *Jack et al., 2001*).

# Materials and methods

**Key resources table**

| Reagent type (species) or resource | Designation | Source or reference | Identifiers | Additional information |
|---|---|---|---|---|
| Gene (*Escherichia coli*) | EmrE | GenBank | Z11877 | |
| Strain, strain background (*Escherichia coli*) | BL21 Gold(DE3) | Agilent Technologies | 230312 | Competent cells |
| Strain, strain background (*Escherichia coli*) | MG1655-Δ*emre* | Creative Biogen | | Deletion of emre from K12 *E. coli* strain MG1655 |
| Recombinant DNA reagent | pWB-EmrE (plasmid) | J. Spreacker, et al., Activating alternative transport modes in a multidrug resistance efflux pump to confer chemical susceptibility. *Nat Commun* 13, 7655 (2022). | | Insertion of gene for EmrE or EmrE mutants (E14Q, Δ107) |
| Recombinant DNA reagent | pET15b-EmrE (plasmid) | Novagen vector pET15b | | Insertion of gene for EmrE or EmrE mutants (E14Q, Δ107) |
| Peptide, recombinant protein | Thrombin (human) | Millipore Sigma | Cat #T7572 | |
| Other | 8-Hydroxypyrene-1,3,6-trisulfonic acid trisodium salt (pyranine) | Millipore Sigma | CAS 6358-69-6 | pH-sensitive dye |
| Chemical compound, drug | Harmane | Millipore Sigma | CAS 486-84-0 | |
| Other | *n*-Decyl-*b*-D-maltopyranoside (decylmaltoside, DM) | Anatrace | Cat #D322 | Detergent |
| Other | 1-Palmitoyl-2-oleoyl-glycero-3-phosphocholine (POPC) | Avanti Polar Lipids | Cat #850457 | Lipid |
| Other | 1-Palmitoyl-2-oleoyl-glycero-3-phosphoglycerol (POPG) | Avanti Polar Lipids | Cat #840457 | Lipid |
| Other | 1,2-Dimyristoyl-*sn*-glycero-3-phosphocholine (DMPC) | Avanti Polar Lipids | Cat #850345 | Lipid |
| Software, algorithm | RAPTOR | http://github.com/uchicago-voth/raptor | commit f17fcc7 | |

## Microplate growth assays

Each EmrE construct was cloned into the pWB vector (*Spreacker et al., 2022*), a low copy number plasmid vector with a p15A origin and pTrc promoter, and transformed into MG1655-Δ*emrE E. coli*. For experiments, LB plates were streaked and grown overnight at 37°C. In the morning, single colonies were picked to inoculate liquid LB cultures at 37°C. Once liquid cultures reached log phase growth, they were diluted back to an OD600 of 0.2 and further diluted 20-fold into microplates with LB media containing the indicated amount of substrate. Growth in microplates at 37°C was monitored for 15 hr using a TECAN Spark or BMG-Labtech microplate reader at OD700 (Ethidium) or OD600. Reported growth curves and final ODs are mean values of two biological replicates containing technical triplicates, with errors calculated using the standard deviation of the mean.

## EmrE expression and purification

Protein expression utilized BL21 (Gold) DE3 *E. coli* transformed with a pET15b plasmid containing the respective EmrE construct, with cells grown in M9 minimal media. Protein was solubilized in decyl maltoside (DM) detergent and purified using immobilized nickel chromatography and size exclusion chromatography as previously described (*Morrison and Henzler-Wildman, 2014*).

### For pyranine fluorescence assays

BL21 Gold (DE3) *E. coli* cells transformed with pET15b-EmrE, pET15b-E14QEmrE, or pET15-Δ107EmrE were grown in M9 minimal media to an OD600 of 0.9. The bacteria were flash cooled and then induced with 0.33 M IPTG overnight at 17°C. The *E. coli* cells were collected with centrifugation, lysed, and the membrane fraction solubilized with 40 mM DM. Purification was via Ni-NTA chromatography followed by cleavage of the N-terminal 6x-His tag using thrombin and then size exclusion chromatography with a Superdex 200 column, with 10 mM decyl maltoside in all buffers (DM, Anatrace, Maumee, OH) as described (*Morrison and Henzler-Wildman, 2014*). Protein concentrations were determined using absorbance at 280 nm with an extinction coefficient of 38,400 l/mol cm (*Morrison et al., 2011*). Fractions containing EmrE in DM were reconstituted into a 3:1 mixture of 1-palmitoyl-2-oleoyl-glycero-3-phosphocholine (POPC, Avanti Polar Lipids, Alabaster, AL) and 1-palmitoyl-2-oleoyl-glycero-3-phosphoglycerol (POPG, Avanti Polar Lipids, Alabaster, AL) liposomes as follows. POPC and POPG in chloroform were dried under nitrogen, washed 3× with pentane to remove residual chloroform, and lyophilized overnight. Dry lipids were hydrated for 1 hr in 100 mM MOPS, 20 mM NaCl, and 1 mM pyranine, pH 6.5, sonicated for 1 min before 0.5% octyl-glucoside was added. The mixture was sonicated for another 30 s and allowed to permeabilize for 15 min at room temperature. Hydrated lipids were mixed with EmrE in DM at a 400:1 lipid:protomer mol:mol ratio (final lipid concentration 12 mg/ml) and allowed to equilibrate for 20 min. Detergent was removed by Biobeads as previously described (*Morrison and Henzler-Wildman, 2012*). Proteoliposomes were extruded 11 times through a 0.2 μm filter (Avanti Polar Lipids, Alabaster, AL) and dialyzed overnight to remove residual pyranine. Proteoliposomes were then concentrated down 10-fold to allow for a final protein concentration of 2 μM upon dilution.

### For SSME transport assays

BL21 Gold (DE3) *E. coli* cells transformed with pET15b-EmrE or pET15- Δ107EmrE were grown in M9 minimal media to an OD600 of 0.9. The bacteria were flash cooled and then induced with 0.33 M IPTG overnight at 17°C. The *E. coli* cells were collected with centrifugation, lysed, and the membrane fraction solubilized with 40 mM DM. Purification was via Ni-NTA chromatography followed by cleavage of the N-terminal 6x-His tag using thrombin and then size exclusion chromatography with a Superdex 200 column, with 10 mM decyl maltoside in all buffers (DM, Anatrace, Maumee, OH) as described13. Protein concentrations were determined using absorbance at 280 nm with an extinction coefficient of 38,400 l/mol cm (*Morrison et al., 2011*). Fractions containing EmrE in DM were reconstituted into POPC (Avanti Polar Lipids, Alabaster, AL) liposomes as follows. POPC in chloroform was dried under nitrogen, washed 3× with pentane, and lyophilized overnight to remove residual chloroform. Dry lipids were hydrated for 1 hr in 50 mM MES, 50 mM MOPS, 50 mM bicine, 100 mM NaCl, and 2 mM MgCl$_2$, pH 7, and permeabilized with 0.5% octyl-glucoside for 15 min at room temperature. Hydrated lipids were mixed with EmrE in DM at a 400:1 lipid:protomer mol:mol ratio (final lipid concentration 2.5 mg/ml) and allowed to equilibrate for 20 min. Detergent was removed by Biobeads as previously described (*Morrison and Henzler-Wildman, 2012*). Proteoliposomes were extruded 11 times through a 0.2-μm filter (Avanti Polar Lipids, Alabaster, AL) and flash frozen in aliquots stored at –80°C until needed for experiments.

### For NMR

Samples for 2D $^1$H-$^{15}$N TROSY experiments, growth was carried out in perdeuterated M9 with $^{15}$N-NH$_4$Cl as the sole nitrogen source, 2H-glucose as the sole carbon source, and 0.5 g/l $^2$H,$^{15}$N isogro. For Δ107-EmrE NMR assignment experiments, growth was carried out in perdeuterated M9 with 1 g $^{15}$NH$_4$Cl, 0.75 g $^2$H,$^{13}$C-glucose, and 0.5 g CND-Isogro per liter. Cells were harvested and EmrE purified in DM as described above. S200 fractions containing EmrE with 10 mM DM were reconstituted

into DMPC (1,2-dimyristoyl-*sn*-glycero-3-phosphocholine, Avanti Polar Lipids, Alabaster, AL) at 75:1 lipid:EmrE monomer mole ratio following the protocol in *Morrison and Henzler-Wildman, 2012* using Biobeads (Bio-Rad Laboratories, Hercules, CA) to remove detergent. EmrE proteoliposomes were collected by ultracentrifugation (100,000 × *g*, 2 hr, 6°C) and resuspended in NMR buffer with DHPC (1,2-dihexanoyl-*sn*-glycero-3-phosphocholine, Avanti Polar Lipids, Alabaster, AL) and freeze–thawed three times to create *q* = 0.33 DMPC/DHPC bicelles (*Bay and Turner, 2009*) (*q* value confirmed with 1D proton NMR). Final NMR samples contained 0.7–1.0 mM EmrE monomer, 10% $D_2O$, 0.05% $NaN_3$, 2 mM TCEP (*tris*(2-carboxyethyl)phosphine), 2 mM EDTA (ethylenediaminetetraacetic acid), and 2 mM DSS (4,4-dimethyl-4-silapentane-1-sulfonic acid) (*Morrison and Henzler-Wildman, 2012*).

## NMR spectroscopy

Triple resonance backbone walk experiments were acquired for backbone assignment of $TPP^+$-bound Δ107-EmrE at pH 5.5 and 45°C using a sample with 1.25 mM $^2H,^{15}N,^{13}C$ Δ107-EmrE and 16 mM $TPP^+$. TROSY-HNCA, TROSY HNcoCA, and TROSY-HNCACB experiments were acquired on a 900 MHz Bruker Avance III NMR spectrometer equipped with a TCI cryoprobe, and TROSY HNCO, TROSY-HNcaCO experiments were acquired on a 750 MHz Bruker Avance III NMR spectrometer equipped with a TCI cryoprobe. Amide assignments were transferred to other pH values using pH titrations. 2D TROSY-HSQC and TROSY-selected ZZ-exchange spectra of Δ107-EmrE at pH 5.5 or 8.5, and $TPP^+$-bound Δ107-EmrE at pH 5.5 or 7.7, were acquired on an 800 MHz Varian VNMRS DD spectrometer equipped with a 5-mm cryoprobe at 45°C using samples with 0.7–1 mM $^2H,^{15}N$ Δ107-EmrE using standard pulse sequences with gradient coherence selection. 70% of the backbone resonances of $TPP^+$-bound Δ107-EmrE were assigned at pH 5.5 by combining standard triple resonance experiments (TROSY-HNCA, TROSY-HNCACB, TROSY-HNCO, and TROSY-HN(CO)CA) with ZZ- exchange data. For NMR pH titrations, identical samples were prepared at the extreme pH values, and the two samples were gradually mixed to create intermediate pH values, ensuring constant protein, lipid, and salt concentrations across the titration.

To analyze the ZZ-exchange experiments, peak intensities were fit using the nlinls function in nmrPipe to accurately extract peak parameters. Residues for analysis were chosen that had all four peaks (two auto peaks, $I_{AA}$ and $I_{BB}$, and two exchange cross-peaks, $I_{AB}$ and $I_{BA}$) resolved in the 2D planes. Exchange with water reduces the peak intensity of the auto and cross-peak from the open face of the transporter at high pH, resulting in greater scatter for the high pH data. The peak intensity ratio was calculated using the method developed by *Miloushev and Palmer, 2005* the Palmer lab:

$$Peak\ intensity\ ratio = \frac{I_{AB}I_{BA}}{I_{BB}I_{AA} - I_{AB}I_{BA}} = k^2t^2 \tag{1}$$

Calculation of this peak ratio cancels out initial peak intensity and intrinsic relaxation rates to first order and depends on the mixing time (*t*) of the ZZ-exchange experiment in a simplified manner as shown in the equation above. Since the forward and reverse rate constants are identical for EmrE in bicelles (*Morrison and Henzler-Wildman, 2014*), there is only a single rate constant for alternating access, *k*.

## Pyranine fluorescence assays

All data were acquired on a TECAN spark instrument. The excitation wavelength was 465 nm (35 nm bandwidth) and the emission wavelength was 530 (25 nm bandwidth). The excitation spectrum maximum of pyranine shifts from 400 to 450 nm as pH increases, so with a constant 465 nm excitation wavelength, the observed fluorescence signal will increase as pH increases. The number of flashes was set to 30 to reduce well-to-well measurement time. To minimize instrument integration time, replicates were allowed to equilibrate for the full 30 min, and an average of the Z-position and gain recorded by the instrument was used as manual input for the reported assays. Liposome stocks with an internal buffer concentration of 100 mM MOPS, 20 mM NaCl, and 1 mM pyranine, pH 6.5. Aliquots were first pipetted into the plate, which was then input into the instrument, and the assay was started to perform instrument checks, at which point the instrument was paused. The plate was ejected, and 198 µl of 100 mM MOPS, 20 mM NaCl pH 7.5 buffer was pipetted into the well containing the liposomes and returned into the instrument to begin recording as soon as possible. Conditions with CCCP contained 1 µl of CCCP at 200 µg/ml on the opposite side of the well for a

final concentration of 1 µg/µl. No gradient conditions were diluted into 198 µl of 100 mM MOPS, 20 mM NaCl pH 6.5 buffer. Reported data are average values of three replicate wells recorded for 30 min each to minimize well-to-well measuring times, with error bars representing the standard deviation of the mean.

## SSME transport assays

All SSME data were acquired on a Nanion SURFE2R N1 instrument. Liposome aliquots were thawed, diluted fourfold, and briefly sonicated. 10 µl of liposomes were added to prepare 3 mm sensors according to a standard protocol (*Thomas et al., 2021*). For comparison of different mutants, sensors were prepared side-by-side for all variants (including all replicates) on the same day using a single batch of sensors to ensure maximum similarity in proteoliposome loading onto the sensor. While results obtained with different batches of sensors prepared on different days show similar results in terms of relative leak between variants, the absolute value varies from batch to batch and day to day. Thus, while Δ107-EmrE was always leakier than WT-EmrE, the absolute flux through the WT- or Δ107-transporter varied between batches of sensors prepared. Data was not averaged or compared across different batches of sensors. Equivalence of the SSME data and pyranine assay demonstrates the success of this approach. Prior to experiments, sensor capacitance and conductance values were obtained to ensure sensor quality. For all experiments, both internal and external buffers contained 50 mM MES, 50 mM MOPS, 50 mM bicine, 100 mM NaCl, and 2 mM MgCl2, with the pH and drug concentration as indicated for each dataset. For data acquisition, sensors were equilibrated with internal buffer, and transport was initiated by perfusion of the external buffer before re-equilibration with the internal buffer. Signals were obtained by integrating the current during perfusion of the external buffer, with the final 100 ms of the initial buffer equilibration used as the baseline. Reported data are average values of data recorded from at least three separate sensors, with error bars representing the standard deviation of the mean.

## pH-detected liposomal transport assays

Liposomal transport assays were performed as previously described (*Robinson et al., 2017*). Briefly, 1 ml aliquots with internal buffer (50 mM MOPS pH 7, 100 mM KCl) were thawed and extruded the day of the experiment as described above. The samples were run over 2 PD-10 spin columns (Cytiva) equilibrated in external buffer (50 µM MES pH 6 with 1 mM KCl and 99 mM NaCl) following the manufacturer's spin protocol. Samples were then diluted to 1.5 ml in external buffer. Eluted samples were added to 2 ml cuvettes with a stir bar, and a microelectrode was inserted and allowed to equilibrate. The pH was monitored in real time by a WINDAQ DI-710 from DataQ at a rate of 100 per second. Aliquots of valinomycin and CCCP at 1 mg/ml in 100% DMSO were thawed and diluted by half in external buffer to better match the pH. During the recordings, valinomycin was added to a final concentration of 1 µg/ml to create a $\Delta\Psi$, harmane to a concentration of 100 µM, CCCP to a concentration of 1 µg/ml as a control, and 50 nmol of HCl was added for quantification.

## Molecular dynamics

All MD simulations were conducted with GROMACS 2020.4 (*Abraham et al., 2015*). Simulation inputs were generated by CHARMM-GUI membrane bilayer builder (*Lee et al., 2016*). The protein was solvated by 162 DMPC molecules, and 40 mM NaCl was added to the water to neutralize the system. The system was coupled to a Nose–Hoover thermostat (*Nosé, 1984*; *Hoover, 1985*) and a Parrinello–Rahman barostat (*Parrinello and Rahman, 1981*), at 310 K and 1 bar, respectively. The system was minimized, then equilibrated with position constraints gradually releasing, as by the default setting of CHARMM-GUI. Then, the system was further equilibrated for 400 ns without constraint, where the RMSD plateaued, and the box sizes were stable. For one simulation, we observed lipids penetrating the protein in this 400 ns equilibration, so we added another 100 ns simulation with backbone constraints to further equilibrate the membrane before releasing these constraints again. Figures were rendered with ChimeraX. The hydrogen distances were analyzed with PLUMED (*The PLUMED consortium, 2019*), by computing the softmin of all hydrogen–hydrogen distances between two side-chains with $\beta$ = 500.

## Water path length calculation

The water path length calculation was implemented in an in-house modification of PLUMED. The algorithm can be briefly described as follows: Each water oxygen is considered a node in a graph, and the distance for each edge connecting two nodes is determined by a function that is close to 1 when the oxygen-oxygen distance is smaller than $r0$ and grows rapidly when it is larger than $r0$. This $r0$ is set to 3 Å, which is the typical distance between the oxygens of hydrogen-bonded water. A more detailed discussion can be found in reference (*Li and Voth, 2021a*). Then, for each frame in the trajectory, the shortest path is found for the graph. We used the two oxygens of E14B at the starting point and the midpoint of Cα of R102A and G57A as the destination.

## Umbrella sampling with MS-RMD

The simulations were run with the LAMMPS MD engine, and the umbrella sampling was carried out as implemented in PLUMED (*The PLUMED consortium, 2019*; *Thompson et al., 2022*). The codes were co-compiled with RAPTOR, a plug-in to model proton transport reactions (*Mordoch et al., 1999*). The source code of RAPTOR is available at https://github.com/uchicago-voth/raptor (*Teng, 2024*). The starting structure was taken from the classical MD simulation of WT-EmrE. A water molecule was protonated at the mouth of the channel, and steered MD was then used to create initial configurations at different CV values. A total of 43 umbrella windows spanning from CV = 0.0 Å to 15.0 Å were used (*Figure 6—figure supplement 1*), with a varying restraint force constant of 80–15 kcal/mol/Å². The CV is defined as

$$x = d_{OC} \cdot e_{PT} \tag{2}$$

where $d_{OC}$ is a vector pointing from the closer glutamate oxygen to the CEC and $e_{PT}$ is a unit vector of the direction of proton transport. Each umbrella window was equilibrated for 1 ns, and then the production run was for 2 ns. The PMF was reconstructed with the weighted histogram analysis method (WHAM, version 2.1.0) (*Grossfield, 2013*). In these simulations, the Cα of the residues at least 10 Å away from the path and those in TM1–TM2 were restrained to its initial coordinate with a 2.4 kcal/mol/Å² harmonic potential to ensure the bias force does not unrealistically distort the protein conformation.

## Acknowledgements

Research reported in this publication was supported by the National Institute of General Medical Sciences of the NIH through grant R01GM053148 (to GAV) and R35GM141748 (to KHW). This study made use of the National Magnetic Resonance Facility at Madison, which is supported by NIH grant R24GM141526 (NIGMS). Computational resources were provided by the Research Computing Center (RCC) at the University of Chicago. M Brousseau was supported in part by the National Institute of General Medical Sciences of the National Institutes of Health under Award Number T32GM008505 (Chemistry–Biology Interface Training Program). The content is solely the responsibility of the authors and does not necessarily represent the official views of the National Institutes of Health.

# Additional information

## Funding

| Funder | Grant reference number | Author |
| --- | --- | --- |
| National Institute of General Medical Sciences | R01GM053148 | Gregory A Voth |
| National Institute of General Medical Sciences | R35GM141748 | Katherine A Henzler-Wildman |
| National Institute of General Medical Sciences | R24GM141526 | Katherine A Henzler-Wildman |
| National Institute of General Medical Sciences | T32GM008505 | Merissa Brousseau |

| Funder | Grant reference number | Author |
|---|---|---|

The funders had no role in study design, data collection, and interpretation, or the decision to submit the work for publication.

## Author contributions

Merissa Brousseau, Conceptualization, Formal analysis, Validation, Investigation, Visualization, Methodology, Writing – original draft, Writing – review and editing; Da Teng, Formal analysis, Validation, Investigation, Visualization, Methodology, Writing – original draft, Writing – review and editing; Nathan E Thomas, Conceptualization, Formal analysis, Supervision, Validation, Investigation, Methodology, Writing – review and editing; Gregory A Voth, Formal analysis, Supervision, Funding acquisition, Methodology, Writing – original draft, Writing – review and editing; Katherine A Henzler-Wildman, Conceptualization, Formal analysis, Supervision, Funding acquisition, Visualization, Methodology, Writing – original draft, Writing – review and editing

## Author ORCIDs

Merissa Brousseau ⬩ https://orcid.org/0000-0003-0555-7177
Da Teng ⬩ https://orcid.org/0009-0000-1905-4277
Nathan E Thomas ⬩ https://orcid.org/0000-0003-3221-6060
Gregory A Voth ⬩ https://orcid.org/0000-0002-3267-6748
Katherine A Henzler-Wildman ⬩ https://orcid.org/0000-0002-5295-2121

Reviewer #1 (Public review): https://doi.org/10.7554/eLife.105525.3.sa1
Reviewer #2 (Public review): https://doi.org/10.7554/eLife.105525.3.sa2
Author response https://doi.org/10.7554/eLife.105525.3.sa3

# Additional files

## Supplementary files

MDAR checklist

## Data availability

All datasets can be found at MendeleyData (https://doi.org/10.17632/28fx2zgvhx.1).

The following dataset was generated:

| Author(s) | Year | Dataset title | Dataset URL | Database and Identifier |
|---|---|---|---|---|
| Brousseau M, Teng D, Thomas N, Voth G, Henzler-Wildman K | 2025 | The C-terminus of the multi-drug efflux pump EmrE prevents proton leak by gating transport | https://doi.org/10.17632/28fx2zgvhx.1 | Mendeley Data, 10.17632/28fx2zgvhx.1 |

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
