## [Editor Report · eLife Assessment]

This study provides a **fundamental** analysis of the EmrE efflux pump, highlighting the role of the C-terminal domain in influencing uncoupled proton leak. The integration of biophysical techniques with molecular dynamics simulations offers **solid** support for the key findings and adds substantial evidence toward a definitive understanding of EmrE transport mechanism.

---

## [Referee Report · Reviewer #1 (Public review)]

Summary:

Work by Brosseau et. al. combines NMR, biochemical assays, and MD simulations to characterize the influence of the C-terminal tail of EmrE, a model multi-drug efflux pump, on proton leak. The authors compare the WT pump to a C-terminal tail deletion, delta_107, finding that the mutant has increased proton leak in proteoliposome assays, shifted pH dependence with a new titratable residue, faster alternating access at high pH values, and reduced growth, consistent with proton leak of the proton motive force.

Strengths:

The work combines thorough experimental analysis of structural, dynamic, and electrochemical properties of the mutant relative to WT proteins. The computational work is well aligned in vision and analysis. Although all questions are not answered, the authors lay out a logical exploration of the possible explanations.

Weaknesses:

A few analyses that were missing in the first submission were included/corrected in the revision.

---

## [Referee Report · Reviewer #2 (Public review)]

Summary:

This manuscript explores the role of the C-terminal tail of EmrE in controlling uncoupled proton flux. Leakage occurs in the wild-type transporter under certain conditions but is amplified in the C-terminal truncation mutant D107. The authors use an impressive combination of growth assays, transport assays, NMR on WT and mutants with and without key substrates, classical MD, and reactive MD to address this problem. Overall, I think that the claims are well supported by the data, but I am most concerned about the reproducibility of the MD data, initial structures used for simulations, and the stochasticity of the water wire formation. These can all be addressed in a revision with more simulations as I point out below. I want to point out that the discussion was very nicely written, and I enjoyed reading the summary of the data and the connection to other studies very much.

Strengths:

The Henzler-Wildman lab is at the forefront of using quantitative experiments to probe the peculiarities in transporter biophysics, and the MD work from the Voth lab complements the experiments quite well. The sheer number of different types of experimental and computational approaches performed here is impressive.

Weaknesses:

The primary weaknesses are related to the reproducibility of the MD results with regard to the formation of water wires in the WT and truncation mutant. This could be resolved with simulations starting from structures built using very different loops and C-terminal tails.

The water wire gates identified in the MD should be tested experimentally with site-directed mutagenesis to determine if those residues do impact leak.

Comments on revisions:

Having reviewed the latest version of the manuscript, I continue to believe that this is a solid paper with important results. I find the new data regarding the computational pKa estimate of E14 compelling.

---

## [Author Response]

The following is the authors’ response to the original reviews.

**Public Reviews:**

**Reviewer #1 (Public review):**

**Summary:**
Work by Brosseau et. al. combines NMR, biochemical assays, and MD simulations to characterize the influence of the C-terminal tail of EmrE, a model multi-drug efflux pump, on proton leak. The authors compare the WT pump to a C-terminal tail deletion, delta_107, finding that the mutant has increased proton leak in proteoliposome assays, shifted pH dependence with a new titratable residue, faster-alternating access at high pH values, and reduced growth, consistent with proton leak of the proton motive force.Strengths:The work combines thorough experimental analysis of structural, dynamic, and electrochemical properties of the mutant relative to WT proteins. The computational work is well aligned in vision and analysis. Although all questions are not answered, the authors lay out a logical exploration of the possible explanations.Weaknesses:There are a few analyses that are missing and important data left out. For example, the relative rate of drug efflux of the mutant should be reported to justify the focus on proton leak. Additionally, the correlation between structural interactions should be directly analyzed and the mutant PMF also analyzed to justify the claims based on hydration alone. Some aspects of the increased dynamics at high pH due to a potential salt bridge are not clear.
**Reviewer #2 (Public review):**
Summary:This manuscript explores the role of the C-terminal tail of EmrE in controlling uncoupled proton flux. Leakage occurs in the wild-type transporter under certain conditions but is amplified in the C-terminal truncation mutant D107. The authors use an impressive combination of growth assays, transport assays, NMR on WT and mutants with and without key substrates, classical MD, and reactive MD to address this problem. Overall, I think that the claims are well supported by the data, but I am most concerned about the reproducibility of the MD data, initial structures used for simulations, and the stochasticity of the water wire formation. These can all be addressed in a revision with more simulations as I point out below. I want to point out that the discussion was very nicely written, and I enjoyed reading the summary of the data and the connection to other studies very much.Strengths:The Henzler-Wildman lab is at the forefront of using quantitative experiments to probe the peculiarities in transporter biophysics, and the MD work from the Voth lab complements the experiments quite well. The sheer number of different types of experimental and computational approaches performed here is impressive.Weaknesses:The primary weaknesses are related to the reproducibility of the MD results with regard to the formation of water wires in the WT and truncation mutant. This could be resolved with simulations starting from structures built using very different loops and C-terminal tails.The water wire gates identified in the MD should be tested experimentally with site-directed mutagenesis to determine if those residues do impact leak.

We appreciate the reviewers thoughtful consideration of our manuscript, and their recognition of the variety of experimental and computational approaches we have brought to bear in probing the very challenging question of uncoupled proton leak through EmrE.

We did record SSME measurements with MeTPP+, a small molecule substrate at two different protein:lipid ratios. These experiments report the rate of net flux when both proton-coupled substrate antiport and substrate-gated proton leak are possible. We will add this data to the revision, including data acquired with different lipid:protein ratio that confirms we are detecting transport rather than binding. In brief, this data shows that the net flux is highly dependent on both proton concentration (pH) and drug-substrate concentration, as predicted by our mechanistic model. This demonstrates that both types of transport contribute to net flux when small molecule substrates are present.

In the absence of drug-substrate, proton leak is the only possible transport pathway. The pyranine assay directly assesses proton leak under these conditions and unambiguously shows faster proton entry into proteoliposomes through the ∆107-EmrE mutant than through WT EmrE, with the rate of proton entry into ∆107-EmrE proteoliposomes matching the rate of proton entry achieved by the protonophore CCCP. We have revised the text to more clearly emphasize how this directly measures proton leak independently of any other type of transport activity. The SSME experiments with a proton gradient only (no small molecule substrate present) provide additional data on shorter timescales that is consistent with the pyranine data. The consistency of the data across multiple LPRs and comparison of transport to proton leak in the SSME assays further strengthens the importance of the C-terminal tail in determining the rate of flux.

None of the current structural models have good resolution (crystallography, EM) or sufficient restraints (NMR) to define the loop and tail conformations sufficiently for comparison with this work. We are in the process of refining an experimental structure of EmrE with better resolution of the loop and tail regions implicated in proton-entry and leak. Direct assessment of structural interactions via mutagenesis is complicated because of the antiparallel homodimer structure of EmrE. Any point mutation necessarily affects both subunits of the dimer, and mutations designed to probe the hydrophobic gate on the more open face of the transporter also have the potential to disrupt closure on the opposite face, particularly in the absence of sufficient resolution in the available structures. Thus, mutagenesis to test specific predicted structural features is deferred until our structure is complete so that we can appropriately interpret the results.

In our simulation setup, the MD results can be considered representative and meaningful for two reasons. First, the C-terminal tail, not present in the prior structure and thus modeled by us, is only 4 residues long. We will show in the revision and detailed response that the system will lose memory of its previous conformation very quickly, such that velocity initialization alone is enough for a diverse starting point. Second, our simulation is more like simulated annealing, starting from a high free energy state to show that, given such random initialization, the tail conformation we get in the end is consistent with what we reported. It is also difficult to sample back-and-forth tail motion within a realistic MD timescale. Therefore, it can be unconclusive to causally infer the allosteric motions with unbiased MD of the wildtype alone. The best viable way is to look at the equilibrium statistics of the most stable states between WT- and ∆107-EmrE and compare the differences.

**Recommendations for the authors:**

**Reviewer #1 (Recommendations for the authors):**
The work is well done and well presented. In my opinion, the authors must address the following questions.(1) It is unclear to a non-SSME-expert, why the net charge translocated in delta_107 is larger than in WT. For such small pH gradients (0.5-1pH unit), it seems that only a few protons would leave the liposome before the internal pH is adjusted to be the same as the external. This number can be estimated given the size of the liposomes. What is it? Once the pH gradient is dissipated, no more net proton transport should be observed. So, why would more protons flow out of the mutant relative to WT?

We appreciate the complexity of both the system and assay and have made revisions to both the main text and SI to address these points more clearly. While we can estimate liposomes size, we cannot easily quantify the number of liposomes on the sensor surface so cannot calculate the amount of charge movement as suggested by the reviewer. We have revised Fig. 3.2 and added additional data at low and high pH with different lipid to protein ratios to distinguish pre-steady state (proton release from the protein) and steady state processes (transport). An extended Fig. 3.2 caption and revised discussion in the main text clarify these points.

We have also revised SI figure 3.2 to include an example of transport driven by an infinite drug gradient. Drug-proton antiport results in net charge build-up in the liposome since two protons will be driven out for every +1 drug transported in. This also creates a pH gradient is created (higher proton concentration outside). The negative inside potential inhibits further antiport of drug. However, both the negative-inside potential and proton gradient will drives protons back into the liposome if there is a leak pathway available. This is clearly visible with a reversal of current negative (antiport) to positive (proton backflow), and the magnitude of this back flow is larger for ∆107-EmrE which lacks the regulatory elements provided by the C-terminal tail. We have amended the main text and SI to include this discussion.

(2) Given the estimated rate of transport, size of liposomes, and pH gradient, how quickly would the SSME liposomes reach pH balance?

Since SSME measurements are due to capacitive coupling and will represent the net charge movement, including pre-steady state contributions, the current values will be incredibly sensitive to individual rates of alternating access, proton and drug on- and off-rates. Time to pH balance would, therefore, differ based on the construct, LPR, absolute pH or drug concentrations as well as the magnitude of the given gradients. For this reason, we necessarily use integrated currents (transported charge over time) when comparing mutants as it reflects kinetic differences inherent to the mutant without over-processing the data, for example, by normalizing to peak currents which would over emphasize certain properties that will differ across mutants. This process allows for qualitative comparisons by subjecting mutants to the same pH and substrate gradients when the same density of transporter construct is present, and care is given to not overstate the importance of the actual quantities of charges that are moving as they will be highly context dependent. This is clearly seen in Fig 3.2 where the current is not zero and the net transported charge is still changing at the end of 1 second. We have amended SI figure 3.2 and the main text to include this discussion.

(3) Given that H110 and E14 would deprotonate when the external pH is elevated above 7 and that these protons would be released to external bulk, the external bulk pH would decrease twice as much for WT compared to delta107. This would decrease the pH gradient for WT relative to the mutant. Can these effects be quantified and accounted for? Would this ostensibly decrease the amount of charge that transfers into the liposomes for WT? How would this impact the current interpretation that the two systems are driven by the same gradient?

The reviewer is correct that there will be differences in deprotonation of WT and ∆107 and the amount of proton release will also change with pH. We have amended Figure 3.2 to clarify this difference and its significance. For the proton gradient only conditions in Figure 3, each set of liposomes were equilibrated to the starting pH by repeated washings and incubation before measurement occurred. For example, for the pH 6.5 inside, pH 7 outside condition, both the inside and outside pH were equilibrated at 6.5, and both E14 residues will be predominantly protonated in WT and ∆107, and H110 will be predominantly protonated in WT-EmrE. Upon application of the external pH 7 solution, protons will be released from the E14 of either construct, with additional proton being released from H110 for WT-EmrE causing a large pre-steady state negative contribution to the signal (Fig. 3.2A). Under this pH condition, we the peak current correlates with the LPR, as this release of protons will depend on density of the transporter. However, we also see that the longer-time decay of the signal correlates with the construct (WT or ∆107) and is relatively independent of LPR, consistent with a transport process rather than a rapid pre-steady state release of protons. Therefore, when we look at the actual transported charge over time, despite the higher contribution of proton release to the WT-EmrE signal, the significant increase in uncoupled proton transport for the C-terminal deletion mutant dominates the signal.

As a contrast, we apply this same analysis to the pH 8 inside, pH 8.5 outside condition where both sets of transports will be deprotonated from the start (Fig. 3.2B). Now the peak currents, decay rates, and transported charge over time are all consistent for a given construct (WT or ∆107). The two LPRs for an individual construct match within error, as the differences in overall charge movement and transported charge over time are independent of pre-steady-state proton release from the transporter at high pH.

(4) A related question, how does the protonation of H110 influence the potential rate of proton transport between the two systems? Does the proton on H110 transfer to E14?

The protonation of H110 will only influence the rate of transport of WT-EmrE as its protonation is required for formation of the hydrogen bonding network that coordinates gating. However, protonation of both E14s will influence the rate of proton transport of both systems as protonation state affects the rate of alternating access which is necessary for proton turnover. This is another reason we use the transported charge over time metric to compare mutants as it allows for a common metric for mutants with altered rates which are present in the same density and under the same gradient conditions. We do not have any evidence to support transfer of proton from H110 to E14, but there is also no evidence to exclude this possibility. We do not discuss this in the manuscript because it would be entirely speculative.

(5) Is the pKa in the simulations (Figure 6B) consistent with the experiment?

We calculated the pKa from this WT PMF and got a pKa of 7.1, which is in close proximity of the experimental value of 6.8

(6) Why isn't the PMF for delta_107 compared to WT to corroborate the prediction that hydration sufficiently alters both the rate and pKa of E14?

We appreciate the reviewer’s suggestion and agree that a direct comparison would be valuable. However, several factors limit the interpretability of such an analysis in this context:

(a) Our data indicate that the primary difference in free energy barriers between WT and Δ107 lies in the hydration step rather than proton transport itself. To fully resolve this, a 2D PMF calculation via 2D umbrella sampling would be required which can be very expensive. Solely looking at the proton transport side of this PMF will not give much difference.

(b) Given this, the aim for us to calculate this PMF is to support our conjecture that the bottleneck for such transport is the hydrophobic gate.

(7) The authors suggest that A61 rotation 'controls the water wire formation' by measuring the distribution of water connectivity (water-water distances via logS) and average distances between A61 and I68/I67. Delta_107 has a larger inter-residue distance (Figure 6A) more probable small log S closer waters connecting E14 and two residues near the top of the protein (Figure 5A). However, it strikes me that looking at average distances and the distribution of log S is not the best way to do this. Why not quantify the correlation between log S and A61 orientation and/or A61-I68/I71 distances as well as their correlation to the proposed tail interactions (D84-R106 interactions) to directly verify the correlation (and suggest causation) of these interactions on the hydration in this region. Additionally, plotting the RMSD or probability of waters below I68 and I171 as a function of A61-I68 distances and/or numbers over time would support the log S analysis.

The reviewer requested that we provide direct correlation analyses between A61 orientation, residue distances (A61-I68/I71), and water connectivity (logS) to better support the claim about water wire formation, rather than relying solely on average distances and distributions.

We appreciate the reviewer’s suggestion to strengthen our analysis with direct correlations. However, due to the slow kinetics of hydration/dehydration events, unbiased simulation timescales do not permit sufficient sampling of multiple transitions to perform statistically robust dynamic correlation analyses. Instead, our approach focuses on equilibrium statistics, which reveal the dominant conformational states of WT- and Δ107-EmrE and provide meaningful insights into shifts in hydration patterns.

(8) It looks like the D84-R106 salt bridge controls this A61-I68 opening. Could this also be quantifiably correlated?

As discussed in response to the previous question, the unbiased simulation timescales do not permit sufficient sampling of multiple transitions to perform statistically robust dynamic correlation analyses.

(9) The NMR results show that alternating access increases in frequency from ~4/s for WT at low and high pH to ~17/s for delta_107 only at high pH. They then go on to analyze potential titration changes in the delta_107 mutant, finding two residues with approximate pKa values of 5.6 and 7.1. The former is assigned to E14, consistent with WT. But the latter is suggested to be either D84, which salt bridges to R106, or the C-terminal carboxylate. If it is D84, why would deprotonation, which would be essential to form the salt bridge, increase the rate of alternating access relative to WT?

We note that the faster alternating access rate was observed for TPP+-bound ∆107-EmrE, not the transporter in the absence of substrate. In the absence of substrate the relatively broad lines preclude quantitative determination of the alternating access rate by NMR making it difficult to judge the validity of the reviewers reasoning. Identification of which residue (D84 or H110) corresponds to the shifted pKa is ultimately of little consequence as this mutant does not reflect the native conditions of the transporter. It is far more important to acknowledge that both R106 and D84 are sensitive to this deprotonation as it indicates these residues are close in space and provides experimental support for the existence of the salt bridge identified in the MD simulations, as discussed in the manuscript.

(10) In a more general sense, can the authors speculate why an efflux pump would evolve this type of secondary gate that can be thrown off by tight binding in the allosteric site such as that demonstrated by Harmane? What potential advantage is there to having a tail-regulated gate?

This was likely a necessity to allow for better coupling as these transporters evolved to be more promiscuous. The C-terminal tail is absent in tightly coupled family members such as Gdx who are specific for a single substrate and have a better-defined transport stoichiometry. We have included this discussion in the main text and are currently investigating this phenomenon further. Those experiments are beyond the scope of the current manuscript.

(11) It is hard to visualize the PT reaction coordinate. Is the e_PT unit vector defined for each window separately based on the initial steered MD pathway? If so, how reliant is the PT pathway on this initial approximate path? Also, how does this position for each window change if/when E14 rotates? This could be checked by plotting the x,y,z distributions for each window and quantifying the overlap between windows in cartesian space. These clouds of distributions could also be plotted in the protein following alignment so the reader can visualize the reaction coordinate. Does the CEC localization ever stray to different, disconnected regions of cartesian phase space that are hidden by the reaction coordinate definition?

The unit vector e_PT is the same across all windows based on unbiased MD. Therefore, the reaction coordinate (a scalar) is the vector from the starting point to the CEC, projected on this unit vector. E14 rotation does not significantly change the window definition a lot unless the CEC is very close to E14, where we found this to be a better CV. For detailed discussions about this CV, especially a comparison between a curvilinear CV, please see J. Am. Chem. Soc. 2018, 140, 48, 16535–16543 “Simulations of the Proton Transport” and its SI Figure S1.In the Supplementary Information, we added figure 6.1 to show the average X, Y, Z coordinates of each umbrella window.

(12) Lastly, perhaps I missed it, but it's unclear if the rate of substrate efflux is also increased in the delta_107 mutant. If this is also increased, then the overall rate of exchange is faster, including proton leak. This would be important to distinguish since the focus now is entirely on proton leaks. I.e., is it only leak or is it overall efflux and leak?

We have amended SI figure 3.2 to include a gradient condition where an infinite drug gradient is created across the liposome. The infinite gradient allows for rapid transport of drug into the liposomes until charge build-up opposes further transport. This peak is at the same time for both LPRs of WT- and ∆107-EmrE suggesting the rate of substrate transport is similar. Differences in the peak heights across LPRs can be attributed to competition between drug and proton for the primary binding site such that more proton will be released for the higher density constructs as described above. This process does also create a proton gradient as drug moving in is coupled to two protons moving out so as charge build-up inhibits further drug movement, the building proton gradient will also begin to drive proton back in which is another example of uncoupled leak. Here, again we see that this back-flow of protons or leak is of greater magnitude for ∆107-EmrE proteoliposomes that for those with WT-EmrE. We have included this discussion in the SI and main text.

Minor(1) Introduction - the authors describe EmrE as a model system for studying the molecular mechanism of proton-coupled transport. This is a rather broad categorization that could include a wide range of phenomena distal from drug transport across membranes or through efflux pumps. I suggest further specifying to not overgeneralize.

We revised to note the context of multidrug efflux.

**Reviewer #2 (Recommendations for the authors):**
Simulations. The initial water wire analysis is based on 4 different 1 ms simulations presented in Figure 5. The 3 WT replicates show similar results for the tail-blocking water wire formation, but the details of the system build and loop/C-terminal tail placement are not clear. It does appear that a single C-terminal tail model was created for all WT replicates. Was there also modeling for any parts of the truncation mutant? Regardless, since these initial placements and uncertainties in the structures may impact the results and subsequent water wire formation, I would like a discussion of how these starting structures impacted the formation or not of wires. I think that another WT replicate should be run starting from a completely new build that places the tail in a different (but hopefully reasonable location). This could be built with any number of tools to generate reasonable starting structures. It's critical to ensure that multiple independent simulations across different initial builds show the same water wire behavior so that we know the results are robust and insensitive to the starting structure and stochastic variation.

We thank Reviewer 2 for their suggestion regarding the discussion of the initial structure. In our simulations, the C-terminal tail was initially modeled in an extended conformation (solvent-exposed) to mimic its disordered state prior to folding. This approach resembles an annealing process, where the system evolves from a higher free-energy state toward equilibrium. Notably, across all three replicas, we observed consistent folding of the tail onto the protein surface, supporting the robustness of this conformational preference.

For the Δ107 truncation mutant, minimal modeling was required, as most experimental structures resolve residues up to S105 or R106. To rigorously assess the influence of the starting configuration, we analyzed the tail’s dynamics using backbone dihedral angle auto- and cross-correlation functions (new Supplementary Figures 10.1 and 10.2). These analyses reveal rapid decay of correlations—consistent with the tail’s short length (5 residues) and high flexibility—indicating that the system "forgets" its initial configuration well within the simulation timescale. Thus, we conclude that our sampling is sufficient to capture equilibrium behavior, independent of the starting structure.

What does the size of the barrier in the PMF (Figure 6B) imply about the rate of proton transfer/leak and can the pKa shift of the acidic residue be estimated with this energy value compared to bulk?

We noticed this point aligns with a related concern raised by Reviewer 1. For a detailed discussion please refer to Point 5 in our response to Reviewer 1.

Experimental validation. The hypotheses generated by this work would be better buttressed if there were some mutation work at the hydrophobic gate (61, 68, 71) to support it. I realize that this may be hard, but it would significantly improve the quality.

Due to the small size of the transporter, any mutagenesis of EmrE should necessarily be accompanied by functional characterization to fully assess the effects of the mutation on rate-limiting steps. We have revised the manuscript to add a discussion of the challenges with analyzing simple point mutants and citing what is known from prior scanning mutagenesis studies of EmrE.